**Data Availability Statement:** All relevant data are within the paper and its Supporting Information files.

# UDP-glucose dehydrogenase (UGDH) activity is suppressed by peroxide and promoted by PDGF in fibroblast-like synoviocytes: Evidence of a redox control mechanism

Ramya Chandrasekaran[1], Colleen Mathieu[2,3], Rishi Sheth[1], Alexandre P. Cheng[3], David Fong[2], Robert McCormack[4], Hani El-Gabalawy[5], Suman Alishetty[1], Mikell Paige[6], Caroline D. Hoemann[1,2,3]*

1 Department of Bioengineering, George Mason University, Manassas, Virginia, United States of America, 2 Institute of Biomedical Engineering, Polytechnique Montréal, Montréal, QC, Canada, 3 Department of Chemical Engineering, Polytechnique Montréal, Montréal, QC, Canada, 4 Department of Orthopedic Surgery, University of British Columbia, Vancouver, BC, Canada, 5 Department of Medicine and Immunology, University of Manitoba, Winnipeg, MB, Canada, 6 Department of Chemistry & Biochemistry, George Mason University, Manassas, Virginia, United States of America

* choemann@gmu.edu

## Abstract

UDP-glucose dehydrogenase (UGDH) generates essential precursors of hyaluronic acid (HA) synthesis, however mechanisms regulating its activity are unclear. We used enzyme histostaining and quantitative image analysis to test whether cytokines that stimulate HA synthesis upregulate UGDH activity. Fibroblast-like synoviocytes (FLS, from N = 6 human donors with knee pain) were cultured, freeze-thawed, and incubated for 1 hour with UDP-glucose, NAD+ and nitroblue tetrazolium (NBT) which allows UGDH to generate NADH, and NADH to reduce NBT to a blue stain. Compared to serum-free medium, FLS treated with PDGF showed 3-fold higher UGDH activity and 6-fold higher HA release, but IL-1beta/TGF-beta1 induced 27-fold higher HA release without enhancing UGDH activity. In selected pro-liferating cells, UGDH activity was lost in the cytosol, but preserved in the nucleus. Cell-free assays led us to discover that diaphorase, a cytosolic enzyme, or glutathione reductase, a nuclear enzyme, was necessary and sufficient for NADH to reduce NBT to a blue formazan dye in a 1-hour timeframe. Primary synovial fibroblasts and transformed A549 fibroblasts showed constitutive diaphorase/GR staining activity that varied according to supplied NADH levels, with relatively stronger UGDH and diaphorase activity in A549 cells. Unilateral knee injury in New Zealand White rabbits (N = 3) stimulated a coordinated increase in synovial membrane UGDH and diaphorase activity, but higher synovial fluid HA in only 2 out of 3 injured joints. UGDH activity (but not diaphorase) was abolished by N-ethyl maleimide, and inhibited by peroxide or UDP-xylose. Our results do not support the hypothesis that UGDH is a rate-liming enzyme for HA synthesis under catabolic inflammatory conditions that can oxidize and inactivate the UGDH active site cysteine. Our novel data suggest a model where UGDH activity is controlled by a redox switch, where intracellular peroxide inactivates, and high glutathione and diaphorase promote UGDH activity by maintaining the active site cysteine in a reduced state, and by recycling NAD+ from NADH.

**Funding:** This work was supported by the Canadian Institutes of Health Research (CIHR; CDH and RM; https://cihr-irsc.gc.ca) under grant #MOP-133729; and George Mason University (https://www.gmu.edu) Start-up funds to CDH. The sponsors played no role in the study design, data collection and analysis decision to publish or preparation of the manuscript.

**Competing interests:** I have read the journal's policy and the authors of this manuscript have the following competing interests: CDH, Scientific Advisory Board and a shareholder of Chitogenx Inc. (formerly Ortho RTi Inc.); RC, CM, RS, AC, DF, RM, HEG, SA and MP have declared that no competing interests exist. This does not alter our adherence to PLOS ONE policies on sharing data and materials.

# Introduction

Fibroblast-like synoviocytes (FLS) line the surfaces of synovial membranes, and serve an important role by releasing hyaluronic acid (HA) to create a viscous, chondroprotective joint fluid. UDP-glucose dehydrogenase (UGDH) is a pleiotropic enzyme that performs a double oxidation of UDP-ɑ-D-glucose (UDP-Glc) to UDP-ɑ-D-glucuronate (UDP-GlcA) while reducing 2 NAD+ (nicotinamide adenine dinucleotide) to 2 NADH [1, 2]. HA synthases require UDP-GlcA to polymerize and release HA to the synovial fluid [3, 4]. HA synthesis is stimulated in various cell types by cytokines, including platelet-derived growth factor-BB (PGDF) [5], interleukin-1β (IL-1β) [6–8], transforming growth factor-β1 (TGF-β1) [5, 9], or the combination of TGF-β1 and IL-1β which was the highest inducer of HA in cultured FLS *in vitro* [10, 11]. The effect of cytokines on UGDH activity however is currently unknown, partly due to the lack of a suitable assay for measuring UGDH activity in cultured cells. The primary purpose of this study was to determine whether cytokines that induce HA synthesis also stimulate UGDH activity. Results from this study could improve our understanding of how endogenous HA levels are regulated in injured or arthritic joints. Data from this study could also provide valuable new ideas on how to utilize platelet-rich plasma (which releases PDGF and TGF-β1) to elicit chondroprotective responses in injured joints [12–14], and improve our understanding of how synoviocytes respond to inflammation.

A method for detecting UGDH activity in cartilage cryosections by enzyme histostaining was originally developed in the 1960s by Balogh and Cohen [15]. Pitsillides et al. [4, 16] subsequently used this method in synovial membrane cryosections, and reported that synovial lining UGDH activity and synovial fluid HA levels were depressed in knee joints with inflammatory joint disease compared to intact knees. These results however are in contradiction with the above-mentioned experiments showing that many cytokines normally found in inflamed joints can stimulate HA production. These contradictory results illustrate a current gap in our understanding of the effect of cytokines on UGDH activity and HA production *in vitro* and *in vivo*. It is currently unclear whether UGDH could be a rate-limiting enzyme in HA synthesis.

In original enzyme staining experiments, unfixed cryosections were incubated with UGDH substrates (UDP-Glc, NAD+), and nitroblue tetrazolium (NBT), where it was assumed that UGDH is the sole NADH-generating system, after which a second enzyme uses NADH to reduce NBT to a blue diformazan dye [15]. However no reports are available on what this second enzyme could be and how it can influence the detection of UGDH enzyme activity in synovial cells. In our study, in addition to modifying the UGDH enzyme histostain to work in cultured FLS, we also sought to gain insights into the second enzyme involved in UGDH enzyme histostaining, and how it can influence UGDH activity detection. During our investigation, we observed that UGDH protein consistently localized to the cytosol, but the UGDH enzyme histostain occasionally localized to the nucleus. These observations led us to carry out cell-free assays which showed that glutathione reductase (GR), an oxidoreductase located in the nucleus of proliferating cells [17, 18], or diaphorase, could both mediate, and were required, for NADH-based reduction of NBT to a formazan dye. Following these insights, we systematically analyzed whether N-ethyl maleimide (NEM, a reported inhibitor of GR and diaphorase), peroxide (an inflammatory mediator) or UDP-xylose (an allosteric inhibitor of UGDH [19]) can inhibit these enzyme staining reactions. We also measured UGDH and "diaphorase" staining activity in A549 lung fibroblasts which are known to express high levels of UGDH protein.

To translate our findings to the context of joint inflammation, we analyzed diaphorase/GR and UGDH activity in synovial membranes collected from *New Zealand White* (NZW) rabbit

knees with and without joint injury. A rabbit model was previously developed in which a unilateral cartilage defect is created, allowed to degenerate for one month, and then treated by microdrilling, which is a bone marrow stimulation cartilage repair therapy [13]. Using this model, we previously observed suprapatellar synovial tissue inflammation at 1-week post-operative that may model inflammatory changes after surgery for articular cartilage repair [20]. In the present study, we analyzed infrapatellar synovium from the same intact and microdrilled chronic defect knees for UGDH and diaphorase/GR activity by enzyme histostaining and compared these results to synovial fluid HA levels. The overall aim of this study was to elucidate the effect of cytokines and joint injury on UGDH activity as it relates to HA release from FLS. Our collective findings support a model in which UGDH activity is favored by a strong reducing environment provided by high GSH levels, whereas inflammation-induced intracellular peroxide could oxidize the UGDH active site cysteine (Cys) and suppress enzyme activity.

## Materials and methods

### Human synovial tissue specimens

All protocols involving humans in research were approved by institutional review boards at Fraser Health Authority (FHREB2014-037), Polytechnique Montreal (CÉR-1415-16), and George Mason University (1480978–1). Small human synovial membrane biopsies were donated by fully informed consenting patients undergoing arthroscopic surgery at the Eagle Ridge Hospital, Port Moody, BC, with mild synovitis and knee pain (N = 6). The treating surgeon (RM) identified appropriate patients satisfying the inclusion criteria (subjects aged 19 to 40, male and female, knee pain due to knee injury of at least 2 on a scale of 0 to 10, 10 being extreme pain, good health, scheduled to undergo knee arthroscopy for knee injury, voluntarily cease anti-inflammatory medication 1 week prior to surgery, are able to provide informed consent, are willing to provide a synovial biopsy), introduced them to the study and provided them with an informed consent letter to take home with them. Patients were then contacted by a member of the hospital research staff, and if the patient verbally agreed to participate, they were scheduled for an appointment to conduct a research questionnaire to review inclusion/exclusion criteria, obtain written informed consent, and relevant patient information. Exclusion criteria included prior local corticosteroid injection in the knee or oral corticosteroids within 1 week, prior treatment with bone-altering medications (teriparatide/PTHrP, bisphosphonates/alendronate, estrogen replacement), HA injections within 6 months prior, chronic inflammatory disorders or poly-arthritis (rheumatoid arthritis), knee infection or other infection, and other criteria (S1 Table). None of the donors had injected or oral corticosteroids in the previous three months. Donor features (gender, age and self-reported pain score on a scale of 0 to 10) were noted prior to surgery, along with synovial joint clinical characteristics. Cartilage ICRS macroscopic score and synovial inflammation (based on the ICRS cartilage injury evaluation package [21]) were evaluated intro-operatively by the surgeon (RM) prior to biopsy collection (Table 1). A small synovial tissue biopsy (~500 mg) was collected under aseptic conditions and placed immediately in Dulbecco's modified Eagle medium (DMEM, 4.5 g/L glucose, product 12100–038, Life Technologies) supplemented with 2 mM L-glutamine, 1.85 g/L sodium bicarbonate, 100 U/mL penicillin/100 µg/mL streptomycin and 50 µg/mL gentamycin (Sigma-Aldrich). Synovial samples were air-shipped from Eagle Ridge hospital on ice packs and FLS were extracted within 24 to 30 hours of collection.

### FLS extraction

FLS were isolated from biopsies according to methods of Rosengren et al. [20, 22]. Any residual blood elements were rinsed with sterile phosphate-buffered saline (PBS) at reception.

**Table 1. Human synovial biopsy donor characteristics (N = 6).**

| Patient Characteristics | Value |
|---|---|
| Gender | N = 3 Male; N = 3 Female |
| Age (median; 25% - 75% IQR; min-max) | 22 years; 19–27 IQR; 19–38 yrs old |
| Body Mass Index (median; 25% - 75% IQR) | 27; 22–32 IQR |
| Pain score (out of 10; mean ± SD) | 5 ± 2 |
| Cartilage ICRS Score (Grade 0, I, II, III, IV; mean) | II |
| Synovial inflammation (0, +, ++, +++; mean) | + (5 out of 6); ++ (1 out of 6) |
| Healthy | Yes (6/6) |
| Athletic | Yes (5/6) |
| Osteoarthritis or rheumatoid arthritis | none |

Biopsies were minced with sterile scissors, placed in a 15 mL conical tube in 5 mL of RPMI (Life technologies, product 31800–022) with Pen/Strep, then combined with 5 mL of 1 mg/mL filter-sterile collagenase type VIII (Sigma-Aldrich, product C2139) in RPMI, incubated for 90 min at 37°C with constant agitation, and passed through a 70-μm nylon mesh strainer (Fisher Scientific). Extracted cells were centrifuged twice at 250×g for 10 min and resuspended each time in 15 mL of DMEM/Pen-Strep/gentamicin with 10% fetal bovine serum (FBS, product 26140–079, Life Technologies). Cells were counted and assessed for viability by trypan blue exclusion (yield: 2.5±0.4 million cells per donor biopsy; cell viability: 48%±11%), and cultured sub-confluent at 37°C and 5% $CO_2$. Medium was changed twice a week and cells were passaged at 90% confluency. FLS were cultured in in DMEM/1% Pen-Strep/0.5% gentamicin/10% FBS and used at a minimum passage 3 (P3) to eliminate type A macrophage-like synoviocytes.

## HA concentration measures

The 24-hour conditioned medium from confluent FLS cultures in 24-well plates (N = 6 donors, seeded at 100,000 cells/well, cultured for 5 days, rinsed and cultured 24h in different culture conditions) was cleared by centrifugation at 180×g for 10 min., then soluble HA was quantified by an enzyme-linked immunosorbent assay that uses aggrecan as a capture reagent and biotinylated aggrecan as the detection reagent (ELISA, product DY3614, R&D Systems). DNA was extracted from cells with guanidine isothiocyanate and quantified by Hoechst fluorescent assay (Tecan M200 Infinity plate reader, Tecan Systems, San Jose, CA, USA) against a standard curve of sheared calf thymus DNA [23]. Data were represented as HA μg/mL/million cells assuming 7.7 pg DNA/cell [23].

## UGDH in situ enzyme stain of in vitro cultured cells

FLS from 5 human donors were cultured in 8-well Permanox LabTek chambers at a seeding density of 40,000 cells per well, for 48 h to reach uniform confluency (FLS from the 6[th] donor yielded only enough primary cells for HA release measures, and pilot UGDH staining). Monolayers were rinsed in serum-free medium (DMEM/Pen-strep/gentamicin) in order to completely remove any serum in the well, then cultured for 24 h in serum-free medium with or without added recombinant human cytokine PDGF-BB (100 ng/mL, product 220-BB, R&D Systems), a combination of IL-1β (1 ng/mL, product 201-LB, R&D Systems) with TGF-β1 (1 ng/mL, product 100-B, R&D Systems), or 10% FBS. After 24 h, the cultured medium from each condition was collected individually and then the FLS monolayers were washed once in PBS. After carefully and completely suctioning out the PBS, the Labteks were incubated for 1–18 h at -80°C. UGDH activity staining solution containing 1 mM β-NAD+ hydrate (0.65

mg/ml, Sigma-Aldrich, product N1511), 5.5 mM UDP-Glc (3.5 mg/ml, Sigma-Aldrich, product U4625), 3.7 mM NBT (3.2 mg/mL, Invitrogen, product N6495), 4.2% (w/v) polyvinyl alcohol (PVA, product 8136, 30,000–70,000 g/mol, Sigma-Aldrich), and 52 mM glycylglycine (Gly-Gly, Sigma-Aldrich, product G1002), was prepared by combining stock solutions of 100 mM Gly-Gly/8% w/v PVA pH 7.8 buffer (PVA was dissolved in 100 mM Gly-Gly pH 7.8 at 60˚C, then degassed for 30 min. under vacuum and purged with $N_2$ gas to remove any oxygen), 10 mg/mL NAD+, 30 mg/mL UDP-glucose and 10 mg/mL NBT (Table 2). For negative controls, UDP-glucose was substituted with ddH$_2$O, or UDP-xylose (Chemilys, GA) was added at 0.2 mM [24] to the UGDH staining solution. After rapid thawing of the monolayers to room temperature, 100 $\mu L$ (96-well plate) or 200 $\mu L$ (Labtek chamber) of the prepared staining solution was applied immediately to monolayers in each well. Samples were stained for 60 min at 37˚C and 5% $CO_2$/20% $O_2$, after which the staining reaction was stopped by washing slides once in ddH$_2$O for 1 to 2 min. Cell nuclei were counterstained with Hoechst 33258, rapidly rinsed in ddH$_2$0, and coverslipped with Aquamount.

## Quantitative histomorphometry of staining intensity

To measure cytokine-dependent UGDH staining, three to six images at 20× magnification (epifluorescence for Hoechst-stained nuclei and bright field with fixed exposure time and light setting for UGDH stain) were collected in the middle and 3 corners of each well with a Zeiss Axiovert 200 microscope, monochrome digital camera and CZI Zeiss software. Pixel intensity

**Table 2. In situ enzyme staining reaction conditions.**

| Staining Solution Reagent | 800 μL solution | Final Concentration |
|---|---|---|
| **UGDH stain** | | |
| 100 mM Gly-Gly, 8% w/v PVA, pH 7.8 | 430 μL | 52 mM GlyGly, 4.2% w/v PVA |
| 50 mM UDP-Glc, disodium (30 mg/mL) | 80 μL | 5 mM (3 mg/mL) |
| 15 mM NAD+ (10 mg/mL) | 50 μL | 1 mM (0.63 mg/mL) |
| 10 mg/mL NBT chloride–vortex well | 240 μL | 3 mg/mL (3.7 mM) |
| **Negative Control** | | |
| 100 mM Gly-Gly, 8% w/v PVA, pH 7.8 | 430 μL | 52 mM GlyGly, 4.2% w/v PVA |
| ddH$_2$0 | 80 μL | — |
| 15 mM NAD+ (10 mg/mL) | 50 μL | 1 mM (0.63 mg/mL) |
| 10 mg/mL NBT chloride–vortex well | 240 μL | 3 mg/mL (3.7 mM) |
| **NADH Reductase ("high" diaphorase)** | | |
| 100 mM Gly-Gly, 8% w/v PVA, pH 7.8 | 430 μL | 52 mM GlyGly, 4.2% w/v PVA |
| ddH$_2$0 | 70 μL | — |
| 14 mM NADH, disodium (10 mg/mL) | 60 μL | 1 mM (0.71 mg/mL) |
| 10 mg/mL NBT chloride–vortex well | 240 μL | 3 mg/mL (3.7 mM) |
| **NADH Reductase ("low" diaphorase)** | | |
| 100 mM Gly-Gly, 8% w/v PVA, pH 7.8 | 430 μL | 50 mM GlyGly, 4.2% w/v PVA |
| ddH$_2$0 | 70 μL | — |
| 1.4 mM NADH, disodium (1 mg/mL) | 60 μL | 0.1 mM (0.071 mg/mL) |
| 10 mg/mL NBT chloride–vortex well | 240 μL | 3 mg/mL (3.7 mM) |
| **Inhibitor added to 800 μL** | | |
| 100 mM N-ethyl maleimide (NEM) | 8 μL | 1 mM |
| 0.3% v/v hydrogen peroxide ($H_2O_2$) | 9 μL | 1 mM |
| 0.03% v/v hydrogen peroxide ($H_2O_2$) | 4.5 μL | 50 μM |
| 2 mM UDP-xylose | 80 μL | 0.2 mM |

per cell was obtained by a blinded observer with ImageJ (https://imagej.nih.gov/ij/) by measuring total pixels per 20× field with fixed threshold settings (1. Image/Adjust/Threshold '0–185', 2. Apply, 3. Analyze/Measure) that completely covered stained cells and did not select unstained areas, and divided by the total number of Hoechst-stained nuclei per field (S1 Fig).

To measure % staining intensity of color images of enzyme-stained cells (cultured FLS or synovial lining cells), 20× images were collected with a digital color camera and Zeiss Axiovert 25 microscope. The polygon tool (ImageJ) was used to crop an area circumscribing 10 to 20 monolayer cells excluding bare areas, or 10 synovial lining cells. Pixels outside the selection were eliminated ("clear outside"). Color threshold was set in RGB (R:0 to 255; G: 0 to 190; B: 0 to 255) to count stained pixels, (R:0 to 255; G: 0 to 254; B: 0 to 255) to measure total pixels, to calculate % purple pixels/total pixels.

## UGDH immunostaining of primary human FLS monolayer cultures

FLS from 5 donors were seeded in 8-well Permanox LabTek Chambers (ThermoFisher, product 177445) at a seeding density of 40,000 cells per well. After 48 h of culture, cells were rinsed thrice in 500 $\mu$L PBS, fixed in 200 $\mu$L ice-cold methanol -20˚C for 5 min), and washed thrice with 500 $\mu$L PBS. As the UGDH antibody supplier warned that the antibody may be sensitive to Triton-X100, fixed monolayers were exposed to 0.1% Triton-X100/PBS solution for 15 min to get ideal permeabilization and then washed thrice with 500 $\mu$L PBS. Triton-X100 was omitted from the remaining steps: 60 min incubation in normal blocking serum from the VECTASTAIN® kit (Vector Labs, product AK 5200) and a 2 h incubation with anti-UGDH rabbit polyclonal antibody (1:50 dilution in 0.3% BSA/PBS, ThermoFisher Scientific, product PA5-57754, 0.1 mg/mL) or anti-vimentin mouse monoclonal antibody (1:200 dilution in 0.3% BSA/PBS of DHSB, clone AMF-17b, 0.31 mg/mL). Negative controls omitted primary antibody. All wells, including negative controls, were rinsed thrice for 5 min in PBS then incubated in biotinylated horse anti-mouse/rabbit IgG (H+L) (Vector Labs, product 30092) from the VECTASTAIN® ABC-AP staining kit as the secondary antibody for 1 h, rinsed with PBS, followed by VECTASTAIN® ABC-AP Reagent for 30 min, then the Vector Red Working Solution (Vector Labs, product SK-5100) and Levamisole (Vector Labs, product 5000–18) prepared according to the manufacture for 30 min in the dark. Cell nuclei were counter-stained with DAPI (1 $\mu$g/mL), rinsed in water, and mounted using Aquamount. Images were acquired at 40× magnification and scale bars added from a NIST-calibrated microscope slide ruler.

## Cell-free kinetic assay

Diaphorase from *Clostridium kluyveri* (Sigma, product D5540, lyophilized powder, 11.1 U/mg) was dissolved in 50 mM Gly-Gly buffer to have a final concentration of 0.35 units/$\mu$L. Glutathione reductase (GR, Sigma, product G9297, in buffered solution containing 25 mM Tris-HCl, pH 7.4, 1 mM EDTA, and 50% (v/v) glycerol) had a concentration of 0.35 units/$\mu$L. Reagents were combined in an Eppendorf tube: 1 mM NADH (Sigma-Aldrich, product N8129) or 1 mM reduced glutathione (GSH, Sigma-Aldrich, product PHR1359), 1 mM NBT in 50 mM Gly-Gly buffer pH 7.8, with or without 1 $\mu$L of diaphorase (0.35 Units) or 1 $\mu$L of GR (0.35 Units), and with or without 1 mM of N-ethyl maleimide (NEM, Sigma, product E1271). After vortex missing, 200 $\mu$L of each reaction was immediately added to a 96-well plate (Olympus, product 25–104) in triplicate, for a plate reader kinetic assay at 37˚C with reading interval of 5 min at 560 nm for 60 min (BioTek Cytation 5). The assay was repeated on 5 distinct occasions.

## Thin Layer Chromatography (TLC) of NADH- or GSH- reduced NBT

Reagents were mixed in microfuge tubes at 1 mM NADH or 1 mM GSH, 1 mM NBT and 50 mM Gly-Gly buffer (pH 7.8), and left to react for 1 h or 24 h. Reduction of 1 mM NBT by 1 mM NADH with or without 0.34 U diaphorase or GR was carried out for 1 h. After the reaction, the tubes were centrifuged at 200×g for 5 min to collect the reduced NBT in the pellet. If a formazan pellet was not obtained, then 10 μL of the soluble reaction mixture was spotted onto a TLC plate. The reduced NBT products from each reaction pellet were transferred to a glass vial and dissolved completely in 40 μL of methanol (Fisher chemical, product A41220), 360 μL of dichloromethane (DCM, Fisher chemical, product D143-4) and 100 μL of tetrahydrofuran (Sigma, product 401757). 10 μL of dissolved diformazan (reduced NBT) from each reaction was spotted on 2 TLC silica gel plates (Glass-Backed Silica, 250 μm, 20x20 cm, F254 plates purchased from Silicycle, product TLG-R10014BK-323) using microcapillary tubes. The first plate analyzed for NADH or GSH reacted with NBT (1 h and 24 h) and controls, and the second plate was analyzed for NADH, with or without diaphorase or GR, reacted with NBT (1 h) and controls. The solvent system for developing the TLC consisted of dichloromethane/methanol 9:1 v/v as the mobile phase in a standard rectangular TLC glass chamber. The spotted TLC plates were placed in the chamber with the solvent. Plates were dried after the run and photographed under visible and ultraviolet light (365 nm, Analytik Jena, Model UVGL-25) to compare the *Rf* value of each spot by measuring the formazan spot migration in a middle lane or near the edge.

## Diaphorase and UGDH enzyme staining without and with NEM inhibitor

FLS from 3 human donors (2 female, 1 male) were seeded in 96-well plates (Falcon, Polystyrene, Tissue culture grade plate, product 353072) at 40,000 cells per well and cultured 2 to 3 days in DMEM+10% FBS. One row of cells was pre-treated for 2 h with 1 mM NEM (50 μL of 100 mM NEM in 5 mL of culture medium, "NEM in vivo"). All medium was aspirated and the cells were quickly washed with 1% w/v PVA/50 mM Gly-Gly pH 7.8 to limit detachment of freeze-thawed cells during the staining. Once the 1% PVA was aspirated, the plate was frozen at -80°C on a metal surface for 1 to 3 h. After thawing the 96-well plate, 4 different staining solutions with NBT were transferred with a multichannel pipette to 3 columns of wells per staining condition: Diaphorase stain with 1 mM NADH or 0.1 mM NADH, UDGH stain with UDP-glucose and NAD+, or negative control stain with NAD+ only (see Table 2). One additional row of cells with all 4 staining solutions was supplemented with 1 mM NEM ("NEM in stain"). The plate was incubated at 37°C and 5% $CO_2$/20% $O_2$ for 60 min after which the staining reaction was stopped by washing the wells once in 200 μL of ddH$_2$O containing Hoechst 33258 for 2 min, followed by a quick wash with 100 μL of PBS. Finally, 50 μL of ddH$_2$O was added to each well and cells imaged at 20× using a Zeiss Axiolab 25 and color digital camera. A total of 4 distinct cultures were analyzed. In other plates A549 cells were cultured and submitted to NADH or UGDH enzyme staining.

## Rabbit synovial cryosection staining for UGDH and diaphorase/reductase

Research involving rabbits was carried out with protocols approved by the University of Montreal Animal Division (Protocol 16–100, Polytechnique Montreal Protocol ANI-1516-20), and exemption to use a narcotic for animal anesthesia (Health Canada, Authorization 41188.12.16), in conformity with the Canadian Council of Animal Care and ARRIVE guidelines [25]. Rabbit infrapatellar synovial tissues were collected post-mortem at necropsy from both knees of N = 3 skeletally mature New Zealand White rabbits from a previous study [20]. Briefly, rabbits were housed in specific pathogen-free environment, one per cage in adjacent

cages to permit sensory contact. The facility had timed light/dark cycle, provided ad libitum food and water and environmental enrichment strategies. Each rabbit had one control intact knee, and a contralateral knee subjected to a small arthrotomy (pre- and post-operative analgesia: fentanyl patch, anesthesia: ketamine-xylazine injection followed by isoflurane/oxygen) to create a chronic cartilage defect that was allowed to degenerate for 28 days, followed by a second small arthrotomy to treat the defect by microdrilling followed by 1 week of *in vivo* repair. Synovial fluid was retrieved by controlled injection and aspiration post-mortem of isotonic saline in each joint and addition of EDTA [20]. Synovial fluid smears and blood workups showed the absence of knee infection. To minimize effects of subjective bias, 1 male and 2 females were used (average body weight 4.2±0.3 kg), with alternating left or right operated knees, and enzyme staining experiments always included the intact and operated knee synovium from a given rabbit. The experimental unit was the knee joint (3 biological replicates). Dissected synovium were placed in a 1:1 v/v solution of 20% sucrose/isotonic saline:Tissue-Tek® optimum cutting temperature compound (OCT, Sakura/Cedarlane) for 2 to 9 h at 4˚C, then frozen over a mixture of acetone and dry ice in OCT and stored frozen at -20˚C to -80˚C. 10-μm thick histological cryosections were prepared and subjected to *in situ* staining for UGDH or diaphorase. UGDH staining was ideally performed within 1 week of cryosection generation. Diaphorase stain with 1 mM NADH and UGDH staining solutions were prepared according to Table 2, also with 1 mM NEM, or 50 μM or 1 mM peroxide (from a 3% v/v solution, Health Tester, NC, USA). Negative controls consisted of NAD+ without UDP-Glc (± 1 mM GSH). Enzyme staining solutions were directly applied to thawed cryosections encircled with a PapPen, and incubated in a humid chamber at 37˚C for 60 or 75 min. Sections were counterstained or not with Hoechst 33258, washed briefly with ddH₂O, mounted with Aquamount and imaged in brightfield with a Zeiss Axiolab 25 microscope, a color digital camera and Zeiss software (CZI). Histomorphometry with fixed thresholding was carried out on a group of 20 to 30 synovial lining cells cropped from 40x magnification images (ImageJ). All stains were repeated at least twice.

## Statistical analyses

Differences between conditions were analyzed by a one-way analysis of the variance (ANOVA) model combined with post-hoc Tukey Honest Significant Differences (HSD) tests using JMP® Pro (v14.1.0, SAS). For differences due to NEM in the enzyme staining solution, the equivalence test treated differences of 0.03 as practically zero. Differences due to knee injury in synovial lining % enzyme stain were analyzed by matched pair design. Significance was set at $p < 0.05$.

## Results

### UGDH immunostaining, enzyme activity and HA synthesis in cultured FLS

Primary human FLS monolayers were cultured under conditions previously associated with low (serum-free), moderate (10% FBS or PDGF) [5] or high (IL-1β+TGF-β1) [11] HA synthetic rates. Cytokines altered the FLS cell morphology to spread (10% FBS), elongated (PDGF) or a mixture of spindle-shaped and rounded (IL-1β+TGF-β1) (Fig 1). Anti-UGDH immunostaining produced a cytosolic stain in all 4 conditions for FLS from 5 different donors (Fig 1, Panel I, A-D). UGDH was also detected in the nucleus in selected serum- or cytokine-stimulated cells (S2 Fig). Negative controls showed no background staining. Vimentin is reported to be expressed by FLS as part of an insoluble intermediate filament network and hence FLS from all the donors were immunostained for vimentin as a positive control. Vimentin cytoskeletal staining was observed throughout the cytoplasm of all FLS in all 5 donors (Fig

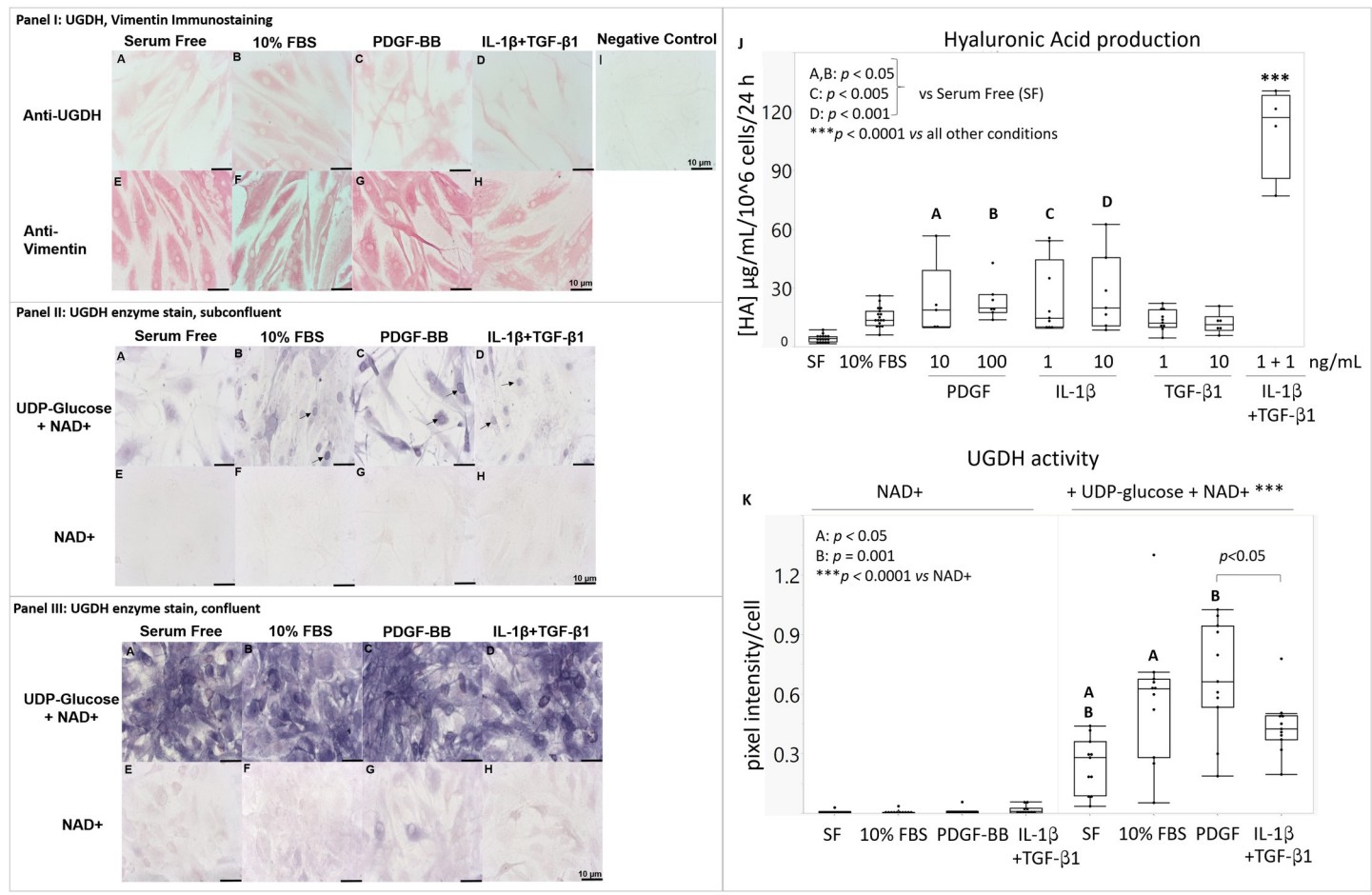

**Fig 1. *In vitro* cultured primary human FLS express UGDH and vimentin protein, and show differential HA synthetic rates and UGDH activity in response to serum and different cytokines.** FLS monolayers were cultured overnight in (A, E) serum-free, (B, F) 10% FBS, (C, G) 100 ng/mL PDGF, or (D, H) 1 ng each IL1β + TGFβ1. (Panel I) Monolayers were fixed and analyzed by immunostaining for (A-D) UGDH protein, (E-H) vimentin protein, or (I) no primary antibody. (Panels II & III) Other monolayers were frozen unfixed then submitted to enzyme histostaining for (A-D) UGDH, and (E-H) NAD+ and NBT as negative controls; representative images show enzyme staining results in areas with (Panel II) low or (Panel III) high cell density. (J) HA synthetic rate was measured after 24 hours of culture in serum-free medium supplemented or not with serum or cytokines (N = 6 donors, 3 male, 3 female). (K) UGDH enzyme activity was measured indirectly by quantitative histomorphometry of staining intensity (N = 5 donors, 2 male, 3 female, 9–12 distinct culture wells per condition). In (E-H, and K), NAD+ without UDP-Glc is the negative control for non-specific staining. Scale bars: 10 μm. Panels J & K show the median, 25% quartile range (box), min-max (whiskers); Significant differences are indicated by letters (A, B, C, D) and ***. For panel K: PDGF was 100 ng/mL, IL1β + TGFβ1 were 1 ng/mL each; SF: serum-free.

1, Panel I, E-H). FLS cultured in serum-free medium produced HA at a rate of 4.1 μg/mL/10$^6$ cells/24 h that was stimulated 6-fold by PDGF or by IL-1β (~25 μg/mL/10$^6$ cells/24h, p<0.05) and 27-fold by IL-1β & TGF-β1 (110 μg/mL/10$^6$ cells/24h, p<0.0001, Fig 1J). These data verified that primary FLS from donors free of chronic joint disease expressed UGDH and responded to cytokines with different HA production rates that were comparable to HA synthetic rates previously measured in cytokine-stimulated OA or RA FLS monolayers [11].

We were aware that all of the UGDH enzyme staining methods prior to our study used unfixed freeze-thawed tissues that we wanted to adapt for monolayer cultures. According to the protocol by Balogh et al. [15] no polyvinyl alcohol (PVA) was used to retain the enzymes in the cells during the 30 to 60 min cytochemical staining of UGDH at pH 8.6, in cartilage sections from mice. In a subsequent study by Mehdizadeh et al [24] mouse knee joints were immersed in 5% PVA before freezing at -70°C and cryosectioning. The cryosections were then exposed to

solutions containing 3 to 6 mg/mL UDP-glucose, 0.8 to 3.2 mg/mL "NAD" and 1 to 3 mg/mL NBT in 15% or 30% PVA, 50 mM Gly-Gly pH 7.8 [24]. Later, Pitsillides et al. [4] measured UGDH activity *in situ* by incubating unfixed frozen rabbit synovial tissue sections at 37˚C in nitrogen atmosphere, in a medium containing 5.3 mM UDP-glucose (3 mg/ml), 0.45 mM "NAD" (0.3 mg/ml), 3.7 mM NBT (3 mg/mL) in 30% (w/v) PVA, 50 mM Gly-Gly buffer pH 7.8. These prior methods did not address the preparation method for 15% or 30% PVA. In our study, PVA solutions over 10% w/v formed a solid gel at room temperature. The data discussed in these prior works also showed a strong background signal in the negative controls that were not supplied with UDP-Glc substrate, for reasons that remain unclear. De Luca et al. [26], Aureli et al. [27], and Rizzotti et al. [28] previously showed histostained monolayer chondrocytes and skin fibroblasts after incubating these cultured cells with the UGDH staining solution of Balogh et al. [15], however negative controls with NBT and no UDP-Glc substrate were missing, and the methods suggested that the staining solution was applied to live cells. We carefully considered all these questions while designing our staining procedure. We reasoned that freezing the monolayers would be necessary to eliminate false-positive signals from mitochondrial NADH oxidases, and as mentioned by Chayen [29], to permeabilize the plasma membrane and allow UDP-Glc, NAD+ and NBT to diffuse into the cells. Cultured cells rinsed with PBS, and left with a very thin layer of PBS on the monolayer before freezing (-80˚C) and thawing (room temperature), followed by rapid application of the PVA staining solution, was found to preserve the cell morphology and sufficiently permeabilize the cells for UGDH activity staining using NBT, NAD$^+$ and UDP-Glc substrate. With our method, we found that 8% PVA in Gly-Gly (pH 7.8) could be prepared and diluted to 4.2% or 5.3% w/v working concentration and prevent UGDH enzyme from diffusing out of the cells [30]. We observed that a hypoxic or nitrogen atmosphere had no influence in staining and it can be carried out at usual 21% O$_2$ atmosphere. With our experiments, we found that glass Labtek slides were not suitable for our colorimetric assay, when Aquamount has to be used for mounting. Most cells cultured in Permanox Labteks or on cell cultured-treated plastic retained their morphology, while cells cultured on soda lime glass Labtek chambers tended to detach. Permanox plastic slides provided a better surface for FLS adhesion after freeze-thaw, and mounting with Aquamount (*unpublished observations*).

UGDH activity was significantly enhanced by 10% FBS (0.59 pixels/cell *vs* 0.24 pixels/cell serum-free, p<0.05) and PDGF (0.64 pixels/cell, p = 0.001 *vs* serum-free) but not by IL-1β +TGF-β1 which elicited variable UGDH activity that was lower than PDGF (0.39 pixels/cell, p = 0.031 *vs* PDGF, Fig 1K). Negative controls without UDP-Glc showed negligible background staining (p<0.0001 *vs* +UDP-Glc, Fig 1K), although a slight background was seen in aggregated cells along the edges of the well along with a stronger UGDH signal (Figs 1 and S1). UGDH staining intensity showed a donor-to-donor variation that tapered off along with HA synthesis in some donor FLS at high passage numbers (P10 to P15, *unpublished observations*). Therefore, only cells from P3 to P9 were used for quantitative histomorphometry measures of UGDH activity. To summarize, PDGF was the only condition tested that stimulated both UGDH activity and HA release compared to serum-free medium.

UGDH protein localized mainly to the cytosol (Figs 1 and S2), and UGDH enzyme staining was mainly cytosolic, however a sub-population of FLS showed an unexpected and distinctly nuclear UGDH enzyme stain, mainly in sub-confluent cytokine-stimulated cells (arrows, Figs 1 Panel II, B-D and S3). This occasional nuclear UGDH enzyme stain was considered pertinent, as other studies have associated nuclear localization of UGDH protein in lung cancer cells with epithelial-to-mesenchymal transition (EMT) and a malignant phenotype [31–33]. This variation in the staining pattern between immunostaining and colorimetric enzyme staining provided a clue that different oxidoreductases could be mediating NBT reduction. According to Balogh et al. [15] when UGDH is used as an NADH-generating system, it is assumed

that diaphorase transfers electrons from NADH to NBT to form a reduced diformazan blue dye. However glutathione reductase (GR) is notably found in the nucleus in dividing cells [17, 18, 34] and could also potentially explain the nuclear stain. To our knowledge, data is lacking as to whether GR can mediate electron transfer from NADH to NBT, or whether reduced glutathione (GSH) could be involved in NBT reduction in the UGDH staining mechanism.

## Diaphorase and glutathione reductase can both mediate NADH-dependent reduction of NBT

Positively charged tetrazolium salts, including NBT, are known to be reduced intracellularly in live cells by NADH-dependent oxidoreductase enzymes [35]. Our in-situ staining reaction, however, is carried out in freeze-thawed (dead) cells. We evaluated whether NADH could spontaneously reduce NBT without an oxidoreductase within the 1-hour time frame of our UGDH staining reaction. In a cell-free assay, 1 mM NADH failed to reduce 1 mM NBT after 30 min at 37°C (Fig 2). Adding diaphorase to the colorless NBT and NADH solution immediately turned the solution to pale brown, due to the instant enzyme-catalyzed reduction of NBT by NADH, producing an OD560 of 0.60 *vs* 0.02 NADH-only after 30 minutes of reaction (p<0.001, Fig 2). Most interestingly, GR also catalyzed NADH reduction of NBT (OD560 of 0.36, p<0.001 *vs* NADH only, Fig 2). GR was partly inhibited by NEM (p<0.01, Fig 2) suggesting that adduct formation in the thiol groups of the GR enzyme limited its ability to mediate electron transfer from NADH to NBT. NEM had no effect on diaphorase-mediated NBT reduction by NADH (Fig 2). Even after one hour of reaction, NADH alone reduced negligible levels of NBT (OD560 of 0.05).

Glutathione transferases were previously identified in human synovium and FLS [36] and under some conditions, GSH was shown to reduce NBT spontaneously *in vitro* [37]. We tested

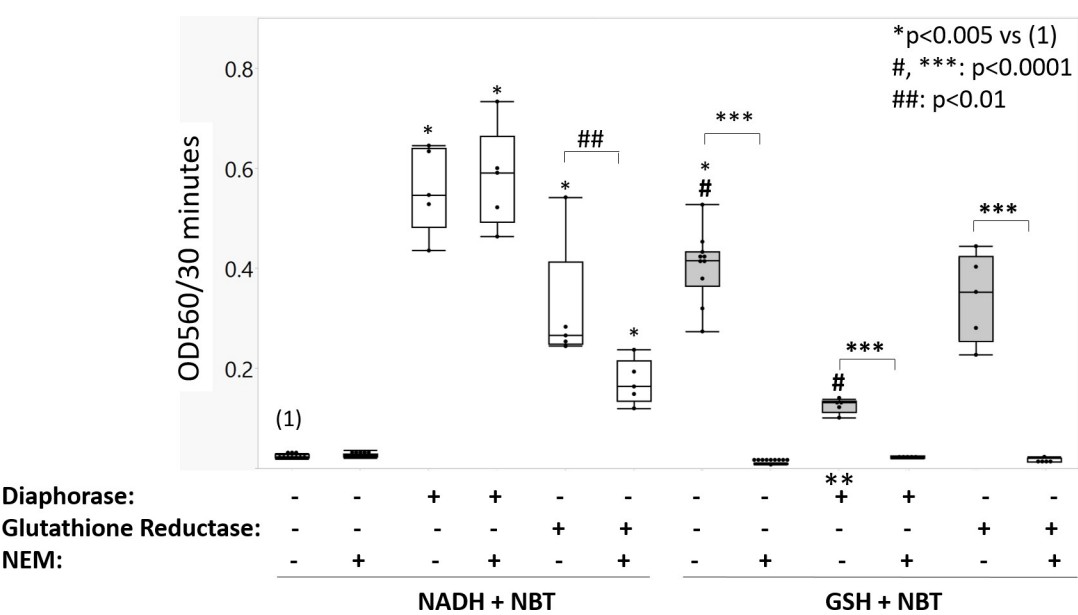

**Fig 2. NADH required diaphorase or GR to reduce NBT whereas GSH spontaneously reduced NBT within 30 minutes of reaction.** Cell-free assays were used to measure the OD560 of reduced NBT absorbance, for mixtures of 1 mM each NADH and NBT (white boxes) or 1 mM each GSH and NBT (grey boxes) after 30 minutes of reaction without or with enzymes or NEM (median, 25% quartile range (box) and min-max (whiskers), N = 3 to 6 independent assays per condition). Symbols: ** evidence that diaphorase can accept electrons from GSH and inhibit GSH from spontaneously reducing NADH (p<0.0001 *vs* no diaphorase); *, ***, #, ##: significant differences between selected conditions.

whether GSH could reduce NBT in a 1 hour timeframe under our enzyme staining conditions. Incubation of equimolar levels of GSH and NBT at 37°C in Gly-Gly buffer pH 7.8 generated an intense blue signal with a mean OD560 of 0.44 after 30 minutes (Fig 2). Unexpectedly, addition of diaphorase to GSH inhibited the NBT reduction reaction (OD560 = 0.16 vs 0.44, GSH-only p<0.0001, Fig 2) whereas addition of GR enzyme had no effect (Fig 2). These data provide novel evidence that diaphorase is capable of directly accepting electrons from GSH. NEM, a thiol-reactive reagent, abrogated GSH-induced NBT reduction in all conditions (p<0.0001, Fig 2).

TLC experiments further confirmed that a 1 hour incubation time is sufficient for 1 mM GSH, but not 1 mM NADH, to spontaneously reduce NBT, although NADH fully reduced NBT after 24 hours (S4 Fig). We also observed using TLC that addition of diaphorase or GR accelerated NADH reduction of NBT to the same diformazan salt as that produced by GSH within 1 h (S4 Fig). To summarize, these data showed that diaphorase or GR was necessary and sufficient to catalyze NADH reduction of NBT to the level of colorimetric detection in a 1-hour UGDH staining reaction, and that GSH reduces NBT in the absence of an oxidoreductase.

## NEM is a potent inhibitor of UGDH activity in freeze-thawed and live FLS

Our cell-free data showed that at least *Clostridium* diaphorase was insensitive to NEM in catalyzing NADH-reduction of NBT; others have reported that NEM interferes with diaphorase activity [39]. We tested whether NEM could inhibit diaphorase activity and therefore prevent UGDH *in situ* staining. In these experiments, UGDH showed cytosolic and occasional nuclear enzyme staining in 3 different FLS donors (2 female, 1 male), and no staining when UDP-Glc was omitted (Fig 3A–3C). FLS were stained in parallel for "diaphorase" by incubating freeze-thawed monolayers with NBT and low NADH (0.1 mM NADH) or high NADH (1 mM, which simulates 100% conversion of 1 mM NAD+ by UGDH). These "diaphorase" staining

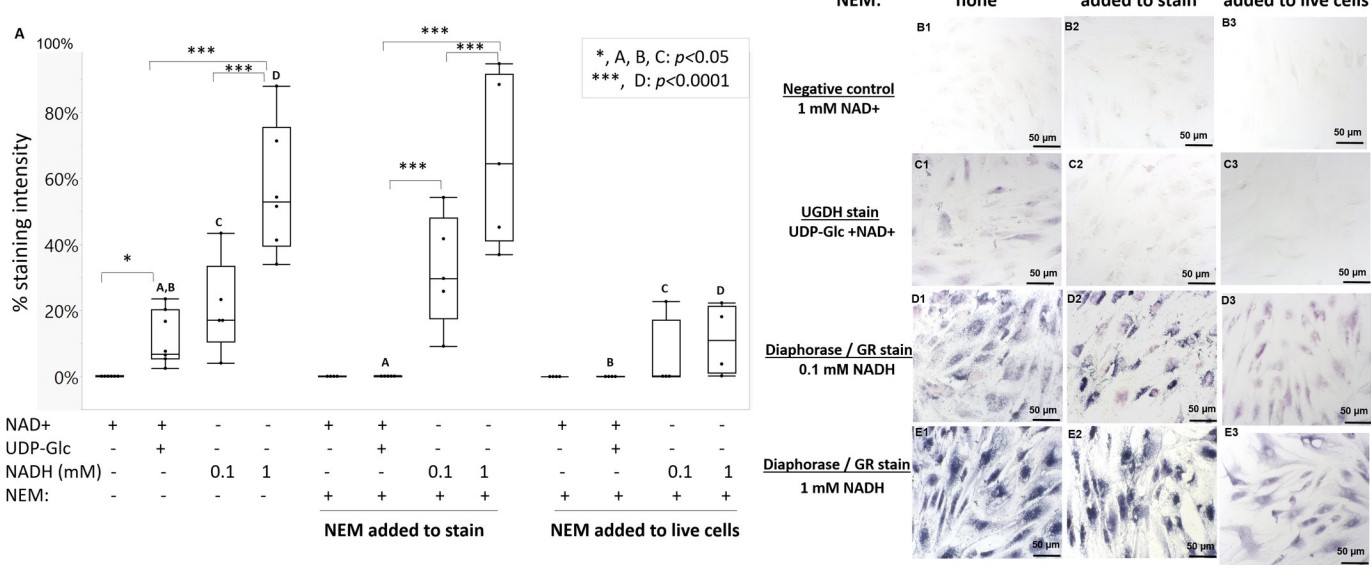

**Fig 3. NEM abolished UGDH enzyme histostaining but only partly inhibited diaphorase staining when added to live cell cultures.** (A) Quantitative histomorphometry of staining intensity for FLS stained for NAD+ only, UGDH, or diaphorase/GR, with and without NEM. NEM was either included in the staining solution ("in stain") or added to cell cultures at 1 mM NEM for 2 hours before enzyme staining ("added to live cells"). Representative images of histostained FLS from one female donor, showed no stain for (B1-B3) the negative control (NAD+ only), NEM-sensitive staining for (C1-C3) UGDH (UDP-Glc and NAD+), and NEM-resistant staining for (D1-D3) "low diaphorase/GR", (E1-E3) "high diaphorase/GR stain"; (B1-E1) no NEM, (B2-E2) NEM "added to stain", or (B3-E3) NEM "added to live cells". In panel A, significant differences are noted with letters and symbols (*, ***). The graph in Panel (A) shows the median, 25% quartile range (box), min-max, distinct cultures with 2 female human donors and 1 male donor FLS (n = 4–7). Scale bars: 50 μm.

conditions produced either a pale (0.1 mM NADH) or an intense dark blue (1 mM NADH) granular and insoluble diformazan in both in the cytosol and nucleus of FLS cells (Fig 3D and 3E, respectively). FLS showed on average 17% or 53% diaphorase staining with low and high NADH, respectively, and 7% UGDH enzyme staining activity (Fig 3A). These results suggested that the UGDH staining reaction had not reached saturation. Adding NEM to the staining solution did not inhibit the diaphorase activity but completely inhibited the UGDH activity ($p < 0.05$, condition 2, Fig 3A). When live cells were pre-treated *in vivo* with 1 mM NEM for 2 h, FLS lost most of the diaphorase activity but completely lost UGDH activity (Fig 3, compare panel C3 with D3-E3). Finally, addition of 0.2 mM UDP-xylose, a well-known allosteric inhibitor of UGDH [19], or 0.05 mM peroxide, to the staining solution strongly suppressed UGDH but not diaphorase staining (0.02 or 0.15 pixels/cell, respectively, vs 1.48 pixels/cell UGDH, $p < 0.01$, S5 Fig).

To verify whether UGDH enzyme staining was specific to FLS, or could be used to detect UGDH activity in other cell types, we carried out enzyme staining in a cell line previously reported to express high levels of UGDH, transformed A549 lung fibroblasts [31, 33]. A549 cells had very strong diaphorase/GR activity (99.8% stain using 0.1 mM NADH), high UGDH activity (93.1% stain) mainly in the cytosol, and a higher background stain (12.1% for NAD + no UDP-Glc, Fig 4) compared to primary human FLS.

### Joint injury at 1 week post-operative shows activated synovial membrane diaphorase and a selective increase in lining cell UGDH activity that is inhibited by peroxidase in a dose-dependent manner

Diaphorase/GR staining activity in rabbit infrapatellar synovial lining cells was 41.7 ± 9.5% in intact knees and 89.0 ± 9.5% in contralateral injured knees at 1 week post-microdrilling ($p < 0.05$, N = 3, Fig 5). Diaphorase/GR activity was also detected in cells in the sub-synovium, and blood vessels of inflamed post-operative synovium (*BV*, Fig 5C4). A faint 4.8 ± 9.5% UGDH staining intensity was detected in intact knee synovial lining cells that was increased in some (but not all) synovial lining cells in injured knees to 17.6 ± 9.5% ($p < 0.05$, N = 3, Figs 5D and S6). Similar to cultured FLS, synovial lining cells stained more strongly for diaphorase/GR than UGDH activity ($p < 0.0001$, Fig 5E). In addition to lining cells, sub-synovial fibroblasts with stellate morphology stained faintly for UGDH. At 1 week post-operative, synovial fluid

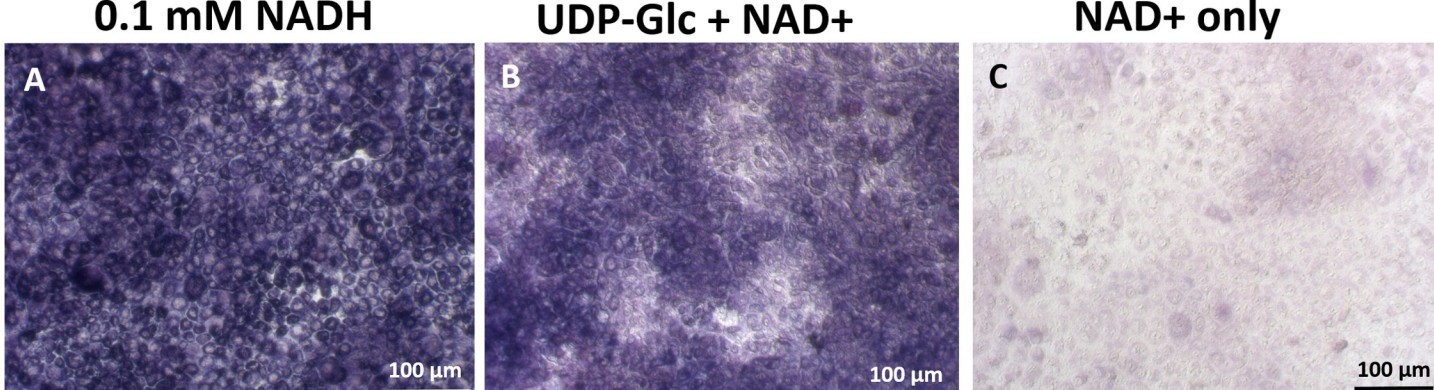

**Fig 4. Intense UGDH and diaphorase enzyme histostaining activity in transformed A549 cells.** By in situ enzyme staining, A549 cells showed relatively high (A) 99.8% diaphorase/GR staining with 0.1 mM NADH, (B) 93.1% UGDH enzyme staining and (C) 12.1% background NBT staining in the absence of UDP-Glc compared to primary human FLS. Scale bars: 100 μm.

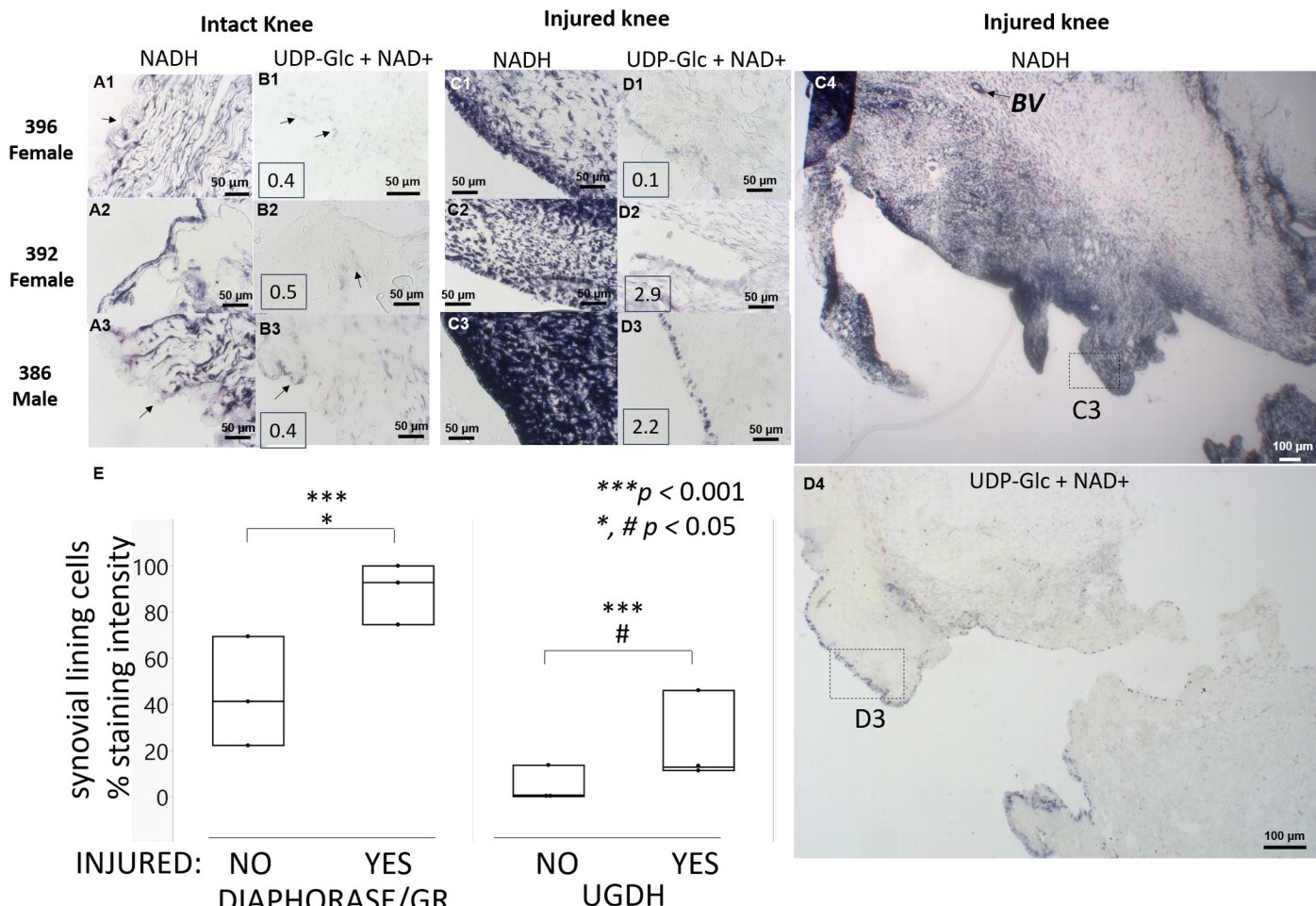

**Fig 5. Injured rabbit knee infrapatellar synovium showed higher diaphorase/GR and UGDH activity, and enhanced synovial fluid HA levels in 2 out of 3 rabbit knees compared to contralateral intact knees.** Panels show 20x magnification images of synovial membrane from regions with maximal UGDH induction, enzyme stained *in situ* for diaphorase (A, C), or UGDH (B, D), in intact (A, B) and contralateral injured knees (C, D). Boxed insets in panels B1-3, D1-3 show measured synovial fluid HA levels (mg/mL). (E) Quantitative histomorphometry of % stain (median, min-max, N = 3). Panels C4, D4 show low-magnification image of male rabbit infrapatellar synovium with dotted boxes showing regions illustrated in panels C3 & D3, respectively. *Symbols and abbreviations*: rabbit ID and sex are shown on the upper left; arrows (A-B) are regions in intact knees where UGDH staining was detected; BV: Blood vessel. Scale bars show 50 μm or 100 μm, as indicated.

HA levels varied from 0.1 to 2.9 mg/mL and did not correlate with UGDH enzyme staining levels (inset, Fig 5B and 5D).

After 60 minutes of enzyme histostaining of synovial membrane cryosections, UGDH activity in synovial lining cells was completely inhibited by 1 mM NEM, 1 mM peroxide, and by 50 μM peroxide (a level of peroxide previously shown to reversibly inactivate the catalytic Cys in protein tyrosine phosphatase 1B, PP1B [40]) (Fig 6). When enzyme staining was extended to 75 minutes, a dose-dependent breakthrough UGDH staining was observed for peroxide-laced stain solutions, but the signal was less intense than UGDH staining without peroxide (Fig 6J and 6N *versus* 6I and 6M). Sections incubated with NAD+ alone (± 1 mM GSH), or UDP-xylose (0.2 mM or 0.05 mM) showed no NBT staining (Fig 6E and 6P, and *unpublished observations)*. "Diaphorase" was strongly detected with or without NEM, peroxide or UDP-xylose (Fig 6 and *data not shown*).

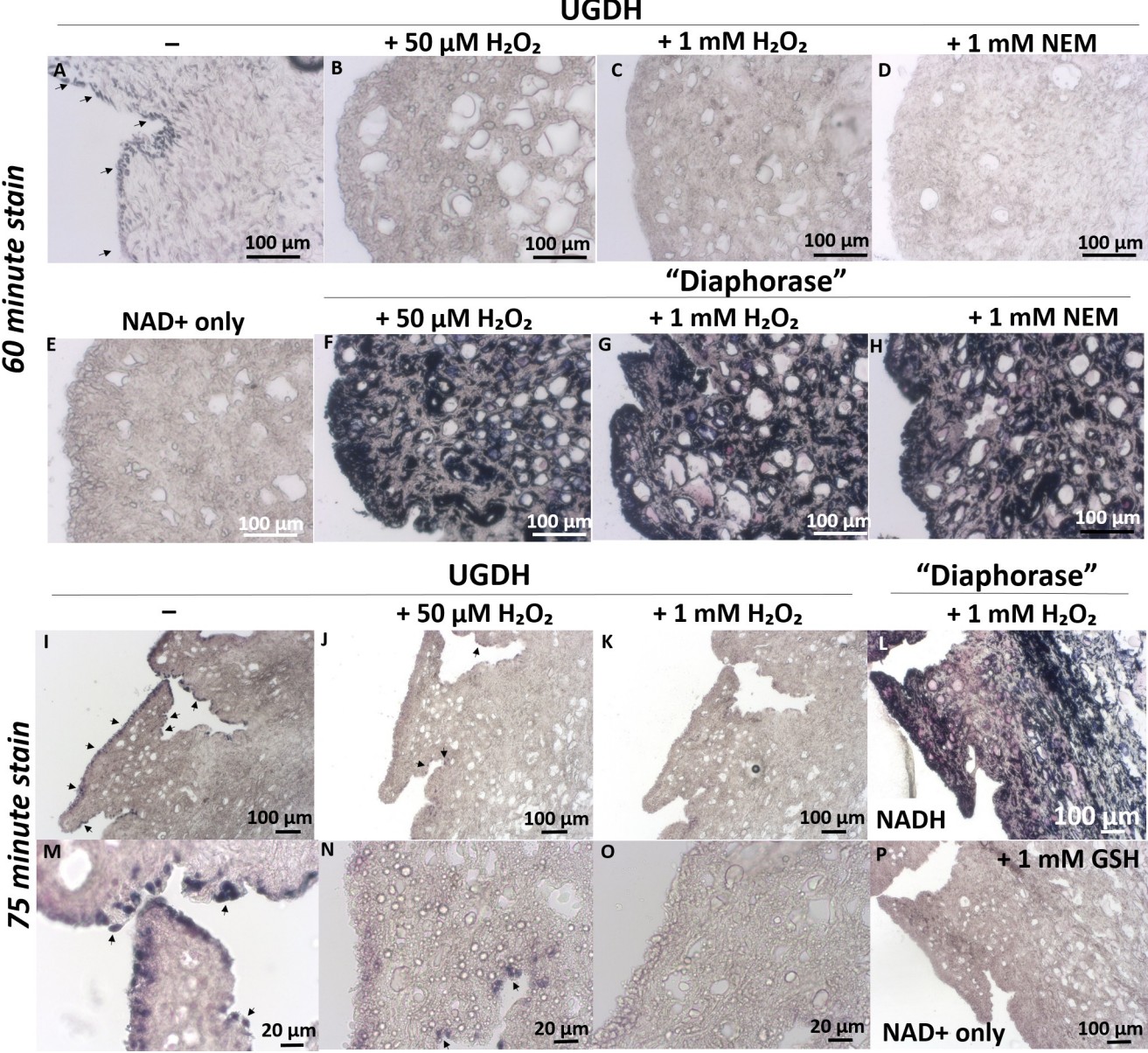

**Fig 6. Peroxide and NEM inhibited UGDH but not diaphorase enzyme staining activity in synovial lining cells from an injured rabbit knee.** In serial synovial cryosections from a New Zealand White male rabbit knee 1 week post-microdrilling, UGDH staining (A, I, M) was inhibited by NEM (D) and by peroxide in a dose-dependent manner (B, C, J, K, N, O) but "diaphorase" was insensitive to thiol modifying inhibitors (F, G, H, L). UGDH and "diaphorase" enzyme histostaining was carried out per standard conditions, in the presence of peroxide or NEM, or with NAD+ only (±1 mM GSH) as indicated, with a 60 or 75 minute incubation at 37°C. UGDH enzyme staining was completely suppressed by NEM (D) and more suppressed by 1 mM peroxide (C, K, O) than 50 μM peroxide (B, J, N). Scale bars are 20 μm or 100 μm, as indicated.

## Discussion

This study is the first, to our knowledge, to present a method for measuring *in situ* UGDH activity in cultured monolayer cells. UGDH enzyme staining was enhanced by PDGF and serum but not inflammatory factors (IL-1β+TGF-β1) in primary human FLS. The anabolic effects of PDGF on UGDH activity and HA release are consistent with the work by Sergijenko *et al.* [41] who showed that murine synovial lining fibroblast-like cells express PDGF receptors

before and after joint injury. These results suggest that platelet factors have the potential to stimulate chondroprotective reactions in FLS. Our results do not support the hypothesis that UGDH activity is rate-limiting for HA synthesis because IL-1β & TGF-β1 stimulated 27-fold higher HA production, potentially through increased hyaluronic acid synthase 2 (HAS2) expression [11, 42], or post-translational HAS activation [43] without proportional increases in UGDH activity.

Our study has certain limitations in that we did not quantify UDP-Glc, UDP-GlcA, or UDP-xylose content in cultured FLS. It is admittedly challenging to measure the levels of these metabolites in biological systems. Others have reported that permeabilized microsomal membranes rapidly lose UDP-sugars [44]. If freeze-thawed FLS monolayers led to UDP-xylose leakage into the cytosol, these hypothetical metabolite levels were insufficient to prevent UGDH activity staining in PDGF- and serum-stimulated FLS. We also cannot rule out the possibility that IL-1β & TGF-β1 selectively upregulated UDP-xylose synthase (UXS) activity and passive transport of UDP-xylose from the endoplasmic reticulum to the cytosol [45]. However in chondrocytes (at least) UXS is subject to end product inhibition by UDP-xylose [46]. We also did not measure UGDH gene expression by Western blot or RT-PCR, but immunostaining showed relatively homogeneous cytosolic UGDH expression and occasional nuclear staining in FLS monolayers from 5 different human donors under all cytokine conditions. UGDH activity was enhanced in rabbit synovial lining at 1-week post-knee injury, but not consistently throughout. These collective results can be reconciled as follows: IL-1β is known to stimulate FLS to produce reactive oxygen species (ROS) [47], which are converted to peroxide in the presence of mitochondrial superoxide dismutase (SOD2) [40]. Peroxide is thus a common intracellular mediator of catabolic inflammation [48]. Moreover, peroxide is an endogenous thiol-reactive compound that like NEM (Fig 3), and other thiol-reactive chemicals [49], could inactivate UGDH by modifying active site Cys 276. It is noteworthy that UDP-xylose inhibits UGDH by masking the active site Cys 276 [50]. A host of enzymes with catalytic Cys (caspase-3, ADH, GAPDH, metalloproteinases, PP1B, dual specificity phosphatase 4, DUSP4) are subject to redox control through oxidation of their active site Cys [40, 48, 51, 52]. PP1B in particular was found to undergo reversible inactivation provided that peroxide levels remained below 1 mM [40]. Our data showed that UGDH activity, which depends on a catalytic Cys [1, 2, 53], is suppressed by 50 μM to 1 mM peroxide (*i.e.*, at levels slightly above the Km 34.4 μM of human UGDH for UDP-Glc [54], Fig 6). Altogether these data suggest that UGDH activity is modulated by a redox switch where oxidative stress depresses enzyme activity and accumulation of GSH helps restore it (Fig 7).

Our study provides novel evidence that UGDH enzyme histostaining is 2-factor mechanism. In our cell-free assay, we observed that 1 mM NADH alone cannot spontaneously reduce NBT effectively after 1 hour, and requires diaphorase or GR to accelerate the reduction. Differences in UGDH activity staining were not due to altered diaphorase/GR activity because freeze-thawed cells showed strong constitutively active diaphorase/GR histostaining, irrespective of the presence of peroxide, NEM, or UDP-xylose. In the optimized *in vitro* enzyme histostain, UGDH enzyme activity can utilize 1 mM NAD$^+$ and produce a maximal 1 mM NADH, which, according to the cell-free assay data, cannot reduce NBT within the 60 min staining reaction, demonstrating that a second enzyme *i.e.* diaphorase and/or GR is necessary. Diaphorase is a cytosolic oxidoreductase that recycles NAD+ from NADH, and can therefore form a coupled reaction or redox hub, that promotes UGDH activity in live cells as well as during the enzyme histostaining reaction (Fig 7). The mismatch in the cytosolic localization of UGDH protein and occasional nuclear enzyme staining led us to investigate if a nuclear enzyme such as GR or intracellular GSH could be involved in addition to diaphorase. GR is known to catalyze the reduction of oxidized GSH (GSSG) to GSH to maintain intracellular GSH levels,

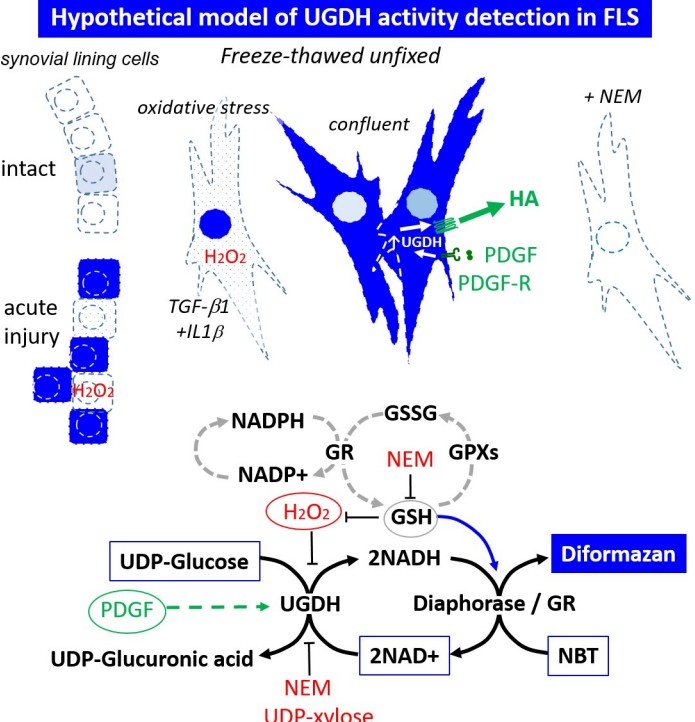

**Fig 7. Hypothetical model representing 2-enzyme UGDH activity staining mechanisms in cultured cells and unfixed cryosections.** Boxed molecules are enzyme staining components, encircled factors and dashed lines represent activities modulated *in vivo* by cytokines. Oxidative stress induced by inflammatory cytokines (TGF-β1+IL-1β) can lead to peroxide generation and reversible inactivation of UGDH unless GSH scavenges the free radicals to sustain UGDH activity. High endogenous GSH can produce background staining by directly reducing NBT and by-passing UGDH activity. *Abbreviations*: PDGF: platelet derived growth factor; GR: glutathione reductase; GSH: glutathione; $H_2O_2$: peroxide; NEM: N-ethyl maleimide (thiol alkylating agent). In the upper panel, Dark blue cells: UGDH-stained cell compartments where cytosolic diaphorase, or nuclear glutathione reductase activity is present; dashed lines: freeze-thaw permeabilized cell membranes.

through an intermediate involving NADPH donation of electrons to FAD (flavin adenine dinucleotide), followed by electron transfer to a disulfide bridge in the GR active center [55]. The electron flow generates a free radical that is then transferred to GSSG, releasing 2 molecules of GSH [56]. Added GR was not necessary for GSH to reduce NBT, but GR catalyzed the reduction of NBT by NADH, enhancing the reaction. This is an unexpected finding and shows that GR can accept electrons from NADH, and has a promiscuous active site that NBT can bind to, permitting its reduction by NADH (Fig 2). Our TLC experiments verified that GR and diaphorase formed the same diformazan product from NADH and NBT. Most interestingly, we found that adding diaphorase enzyme could impair the GSH-induced reduction of NBT which is again a novel finding. This suggests that if high diaphorase is present in the FLS, it could impair the direct reduction of NBT by the intracellular GSH, making diaphorase the most effective secondary factor involved in detecting UGDH activity by reduction of NBT. In case of NADH, NEM did not inhibit NBT reduction by NADH alone or in the presence of diaphorase, but partly inhibited the GR-catalyzed NBT reduction by NADH. This latter result was expected as NEM has been reported to inhibit GR through sulfhydryl alkylation preventing enzymatic catalysis through its active site.

The UGDH active site contains Cys276 which is implicated in the second oxidation step of UDP-Glc through formation of a thiohemiacetal with the C6 carbon, after which NAD$^+$

accepts a hydride followed by release of a UDP-GlcA and a second NADH molecule [1, 2, 53]. It was previously noted that reaction of cyanide with Cys276 inactivates UGDH [2]. We now report that NEM inactivates UGDH activity, most probably by forming an adduct with Cys276 preventing thiohemiacetal formation. Peroxide suppressed but did not completely abrogate UGDH activity, which could be explained by a reversible oxidation of the catalytic Cys276 similar to the mechanism described for PP1B and DUSP4 [40, 48, 52]. According to cryostallographic data of Salmeen et al [40], exposure of PP1B to peroxide at concentrations below 1 mM leads to active site Cys oxidation to form sufenic acid that then reacts with a near-by amino acid to form a reversible sulphenyl-amide bond along with substrate trapping. Other active site Cys enzymes have distinct redox regulatory mechanisms [51] and the precise mechanisms of UGDH redox regulation remain to be identified. GR is known to localize to the nucleus [17] and by producing GSH helps protect DNA from ROS. A faint nuclear signal was present in FLS immunostained for UGDH which could be due to low levels of nuclear UGDH. The distinct nuclear UGDH enzyme stain we observed could therefore occur through peroxide-depressed cytosolic UGDH activity, and nuclear or perinuclear UGDH activity allowing GR reduction of NBT by NADH. Given that GSH can directly reduce NBT *in vitro*, this leads us to hypothesize that accumulation of GSH could produce a nonspecific background stain (Fig 7). It is notable that A549 cells were reported to express high levels of GSH (8.3±2.2 mM) [57], which could explain the higher background staining in these cells. High GSH levels could also suppress peroxide from switching off UGDH activity under inflammatory states.

GR holds high relevance when it comes to chronic oxidative stress in the RA synovium. GR activity is significantly increased in RA synovium but not sufficiently enough to protect the GAGs in the synovial fluid, letting the ROS to attack the knee joint, leading to the destruction of cartilage [58]. Diaphorase on the other hand is cytosolic, found in synoviocytes, FLS, fibroblasts and inflammatory cells in the RA synovium [59, 60]. This was consistent with our results from staining the rabbit sections for diaphorase and UGDH activity. Pitsillides et al. [16] previously argued that non-inflamed synovial lining cells have strong UGDH activity compared to rheumatoid arthritis synovium. Our results in rabbit synovial sections contradict these results where UGDH was barely detectable in intact knees, and upregulated with diaphorase in injured knee synovial lining. Potential explanations for the discrepancies between our results and that of Pitsillides is that the "normal" synovium from these former studies used cancer patient joints, and rheumatoid synovial lining cells may have inactivated UGDH due to intracellular ROS or peroxide. As for cancer patient synovium, UGDH could be upregulated to serve another role such as supplying UDP-GlcA for glucuronidation and clearance of cancer medications [61]. One limitation of the enzyme histostaining approach is that a negative result cannot distinguish between lack of protein expression or suppressed enzyme activity. For this reason, we recommend that UGDH enzyme histostaining experiments include parallel staining of serial sections for diaphorase activity as a control for cell viability and inflammatory status.

The rabbit model has certain limitations as a model of human knee inflammation, including limited motility in a caged environment, a higher metabolic rate, and scant joint fluid. The infrapatellar synovium was selected as a tissue of interest for its proximity to the chronic microdrilled defect and may not be representative of the entire synovial membrane in the joint. Even with these shortcomings, the unilateral rabbit model affords a unique chance to carry out a side-by-side comparison of intact knee and injured knee synovial tissue responses at a fixed time point, compared to human subject research where synovial fluid and tissues are usually only analyzed from a single injured or diseased joint at an uncontrolled time point in disease progression. Inbred rabbits are also free of comorbidities, environmental conditions, medications, and prior intra-articular treatments that can confound data interpretation. Our *in vivo* results were limited by low sample number but adequate to show that UGDH activity

can be stimulated by joint injury, suppressed by trace levels of peroxide, and accompanied by widespread diaphorase/GR activation.

Our data have ramifications in the work by Wang et al. [31], who identified UDP-Glc as having a type of tumor-suppressor role in A549 cells. According to their model, the metabolite UDP-Glc induces an allosteric shift in Hu antigen R (HuR), which prevents HuR from forming a stabilizing complex with the 3' untranslated region of *SNAI1*, the mRNA encoding SNAIL [31]. Also according to their model, EGF stimulation of A549 cells induces the phosphorylation and association of UGDH with HuR, which was proposed to deplete UDP-Glc, and permit a stable HuR:*SNAI1* interaction. The SNAIL transcription factor is then expressed and migrates to the nucleus where it drives EMT by suppressing E-cadherin and promoting vimentin expression and cell motility. Wang et al. [31], localized UGDH protein to the A549 nucleus, although we observed UGDH enzyme staining in the A549 cytosol and nucleus, suggesting that complex formation between UGDH and HuR may not be required for bioactive depletion of UDP-Glc. UGDH activity was implicated to-date in lung adenocarcinoma, breast cancer and glioblastoma [31–33, 62]. Our data additionally implicate diaphorase, GR, peroxide, and GSH in this process. Our findings provide potential explanations for the paradoxical metastasis-promoting effects of anti-oxidants [63, 64]. The UGDH and diaphorase enzyme staining technique described here could be readily applied to various cancer cell lines, and unfixed frozen cryosections from tumors, to provide information on cell types and subcellular location in which UGDH activity is enhanced. Furthermore, data from Pitsillides et al. [16] showed displacement of UGDH + rheumatoid synovial lining cells to the sub-synovium, and our experiments revealed signs of selected UGDH+ FLS motility in the injured synovium (S6 Fig) and UGDH activation by PDGF. The idea that UGDH activity could be involved in "tumor-like" changes observed in rheumatoid synovium [65, 66] is a hypothesis that warrants further investigation.

## Conclusions

We report a new method to detect *in situ* UGDH activity in cultured cells, through a 2-enzyme NBT-based histostaining reaction that requires both UGDH and endogenous diaphorase or GR for colorimetric detection. The staining method can be used to measure UGDH activity in cultured cells, and to screen specific factors for their ability to promote or inhibit UGDH activity. Diaphorase in collaboration with GR protect the cell from oxidative stress, and together could promote UGDH activity *in vivo* by recycling NAD+ from NADH and maintaining the catalytic Cys in a reduced and active state. Based on this study, UGDH *in situ* enzyme staining cannot be used as an accurate predictor of FLS HA production, or as a method to label synovial lining cells in non-inflamed synovial membranes. Instead, our study revealed that UGDH activity can be promoted by PDGF, and controlled by the cellular redox state in a potentially reversible manner.

## Supporting information

**S1 Fig. Representative images used for quantitative histomorphometry of UGDH enzyme histostaining intensity (UDP-Glc + NAD+) and background staining (NAD+).** UGDH staining was higher at the culture well edge, where cells were aggregated compared to the middle of the well, therefore fields from the edge and middle of the well were averaged for the final quantitative result per culture well. Panels (A1-P1) show UGDH staining (grey scale) and (A2-P2) show Hoescht-stained nuclei of the same field. Scale bars: 50 μm.
(TIF)

**S2 Fig. UGDH protein showed detectable nuclear staining in selected cells.** (A-D) Phase contrast and (E-H) matching epifluorescent images of UGDH immunostain and (I-L) DAPI-

counterstained fluorescent nuclei (I-L) of FLS (P4) extracted from a synovial biopsy from a 23-year old male donor and cultured under different cytokine conditions as indicated, fixed in acetone, and immunostained for UGDH with red substrate detection which is fluorescent. Symbols: small white arrows: cytosolic UGDH immunostain; open arrows: detectable nuclear UGDH immunostain. Scale bars: 20 μm.
(TIF)

**S3 Fig. Nuclear UGDH enzyme histostaining was observed in selected FLS cells.** (A, B) UGDH enzyme histostaining was occasionally detected in the nuclear or perinuclear compartment in sub-confluent cells in the middle of the culture well, and (C, D) was mainly cytosolic in more confluent cells at the edge of the well. (A, C) bright field and (B, D) matching epifluorescence image of the same field showing Hoechst-stained nuclei of FLS stimulated here with 1 ng/mL each IL1β + TGF-β1 and stained for UGDH enzyme activity. Scale bars: 20 μm.
(TIF)

**S4 Fig. TLC demonstration that NADH and GSH reduce NBT to the same formazan product with different kinetics.** Arrowheads show the origin and final solvent front; UV light was absorbed in the TLC plate by NADH and NBT at the origin, and by diformazan at the arrow. Reduced formazan had an *Rf* value ranging from 0.42 to 0.44 [38] depending on whether the lane was in the middle or edge of the TLC plate. In the absence of enzyme, GSH reduced NBT after 60 minutes but NADH only reduced NBT after 24 hours. Diaphorase or GR was necessary and sufficient for NADH to reduce NBT within 1 hour. See (S1 File) for original TLC plate images.
(TIF)

**S5 Fig. UDP-xylose and peroxide inhibited UGDH but not diaphorase staining activity in cultured monolayer FLS.** Cells were from a 23-year old male donor (passage 4). FLS stained for UGDH (A: bright field; B: phase contrast) or UGDH with UDP-xylose (C: bright field; D: phase contrast), diaphorase (E), or diaphorase with UDP-xylose (F). In panel (G), quantitative histomorphometry was used to measure the relative staining intensity in the absence or presence of UDP-xylose or peroxide. *Symbols*: $^*$p<0.01 *vs* UGDH; $^{**}$ p<0.0001 vs UGDH. Scale bars: 50 μm.
(TIF)

**S6 Fig. Most synovial lining cells in injured rabbit knees expressed diaphorase/GR, and some, but not all, synovial lining cells expressed UGDH activity.** Serial infrapatellar synovial membrane cryosections from male rabbit 386 were enzyme histostained for (A) diaphorase/ GR, and (B) UGDH activity. Note that some UGDH+ cells are displaced to the sub-synovium.
(TIF)

**S1 Table. Inclusion/exclusion criteria for human subjects recruited to the study.**
(DOCX)

**S1 File. Supporting information for S4 Fig including lane contents and original TLC images.**
(PDF)

## Acknowledgments

We thank Julie Tremblay for Quality Assurance.

## Author Contributions

**Conceptualization:** Ramya Chandrasekaran, Robert McCormack, Hani El-Gabalawy, Caroline D. Hoemann.

**Data curation:** Ramya Chandrasekaran, Colleen Mathieu, Caroline D. Hoemann.

**Formal analysis:** Ramya Chandrasekaran, Colleen Mathieu, Rishi Sheth, Alexandre P. Cheng, Suman Alishetty, Mikell Paige, Caroline D. Hoemann.

**Funding acquisition:** Robert McCormack, Caroline D. Hoemann.

**Investigation:** Ramya Chandrasekaran, Colleen Mathieu, Rishi Sheth, Alexandre P. Cheng, David Fong, Robert McCormack, Suman Alishetty, Caroline D. Hoemann.

**Methodology:** Ramya Chandrasekaran, Colleen Mathieu, Rishi Sheth, Alexandre P. Cheng, David Fong, Hani El-Gabalawy, Suman Alishetty, Mikell Paige, Caroline D. Hoemann.

**Project administration:** Robert McCormack, Caroline D. Hoemann.

**Resources:** Robert McCormack, Mikell Paige, Caroline D. Hoemann.

**Supervision:** Robert McCormack, Mikell Paige, Caroline D. Hoemann.

**Validation:** Ramya Chandrasekaran, David Fong, Suman Alishetty, Caroline D. Hoemann.

**Visualization:** Ramya Chandrasekaran, Caroline D. Hoemann.

**Writing – original draft:** Ramya Chandrasekaran, Caroline D. Hoemann.

**Writing – review & editing:** Ramya Chandrasekaran, Colleen Mathieu, Alexandre P. Cheng, David Fong, Robert McCormack, Hani El-Gabalawy, Suman Alishetty, Mikell Paige, Caroline D. Hoemann.

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
