## [Decision Letter · Decision Letter 0]

27 Jun 2022

PONE-D-22-03359UDP-glucose dehydrogenase (UGDH) activity is suppressed by peroxide and promoted by PDGF, diaphorase and glutathione reductase in fibroblast-like synoviocytesPLOS ONE

Dear Dr. Hoemann,

Thank you for submitting your manuscript to PLOS ONE. After careful consideration, we feel that it has merit but does not fully meet PLOS ONE’s publication criteria as it currently stands. Therefore, we invite you to submit a revised version of the manuscript that addresses the points raised during the review process.

We look forward to receiving your revised manuscript.

Kind regards,

Oreste Gualillo, PharmD, PhD

Academic Editor

PLOS ONE

“I have read the journal's policy and the authors of this manuscript have the following competing interests: CDH, Scientific Advisory Board and a shareholder of Ortho RTi; RC, CM, RS, AC, DF, RM, HEG, SA and MP have declared that no competing interests exist.”

Additional Editor Comments:

Dear Dr. Hoemann, after careful review of your article, I decide for a Major Revision of your article. I suggest to follow all the points and recommendations raised by both reviewers , in particular those suggested by the reviewer 2. After your pertinent modifications of the manuscript, the paper will be resubmitted to reviewers for final decision. Please, if you need extra time for revision, let me know.

Oreste Gualillo, PharmD, PhD

Reviewers' comments:

Reviewer's Responses to Questions

**Comments to the Author**

1. Is the manuscript technically sound, and do the data support the conclusions?

Reviewer #1: Yes

Reviewer #2: No

2. Has the statistical analysis been performed appropriately and rigorously? 

Reviewer #1: Yes

Reviewer #2: No

3. Have the authors made all data underlying the findings in their manuscript fully available?

Reviewer #1: Yes

Reviewer #2: No

4. Is the manuscript presented in an intelligible fashion and written in standard English?

Reviewer #1: Yes

Reviewer #2: Yes

5. Review Comments to the Author

Reviewer #1: Hyaluronic acid is a polymer of enormous biological interest, and its concentration plays a decisive role in tissues both chemically and physiologically. This manuscript presents a histochemical method to measure the activity of the enzyme UGDH that synthesizes an HA precursor (UDPGlcUA). In the course of optimization, however, the authors were able to add an important piece to better understand the regulation of this enzyme in cellular and tissue models. The presented study aims to test whether stimulation of HA synthesis in cells involves a key enzyme for the synthesis of the HA precursor (UDPglucuronic) UGDH. UGDH has unique characteristics, for example, it is the only enzyme that can perform double oxidation on the same carbon, a unique feature in mammals.

The data do not support the hypothesis that UGDH is a rate-liming enzyme for HA synthesis under catabolic inflammatory conditions that can oxidize and inactivate the UGDH active site cysteine. In this way, the work better defines the differences between stimulation by PDGF and by cytokines, which act on the UGDH enzyme in completely different ways.

PDGF stimulates HA production and UGDH activity in primary human FLS whereas inflammatory cytokines stimulated dramatic HA production but without significant increases in UGDH enzyme activity.

The data are obtained by using diaphorase and glutathione reductase (GR) and showed that diaphorase or GR was necessary and sufficient to catalyze NADH reduction of formazan (NBT) to the level of colorimetric detection in a one-hour UGDH staining reaction.

The data are intriguing and the text well written, the text is clear and the figures well done. The literature is properly described.

There are few concerns that can be considered:

1) in line 563 the authors report “This study is the first, to our knowledge, to present a method for measuring in situ UGDH activity in cultured monolayer cells” but really there are a couple of papers addressing the question, in particular the positivity of the UGDH at the edge of cell cultures:

Castellani AA, De Luca G, Rindi S, Salvini R, Tira ME. Regulatory mechanisms of UDP-glucuronic acid biosynthesis in cultured human skin fibroblasts. Ital J Biochem. 1986 Sep-Oct;35(5):296-303. PMID: 3804697.

Rizzotti M, Cambiaghi D, Gandolfi F, Rindi S, Salvini R, De Luca G. The effect of extracellular matrix modifications on UDP-glucose dehydrogenase activity in cultured human skin fibroblasts. Basic Appl Histochem. 1986;30(1):85-92. PMID: 3718422

Nevertheless these papers are old and published in Journals not readily available and the techniques here reported are clearly updated.

2) the glucuronic acid is also a substrate for other glycosaminoglycans, and the authors did not mention it. Moreover it may be interesting to underline if the stimulation with PDGF ot TGFbeta and IL-1 alters the other GAG synthesis. Basically, is there a competition for the substrate in these cells? More HA is coupled with minor synthesis of other GAGs in Golgi? In these cells the carriers carrying UDPsugars in Golgi have very low Km in order to preserve GAG synthesis?

3) This study proposes new data and highlights that UGDH activity is controlled by a redox switch, where intracellular peroxide inactivates, and high glutathione and diaphorase promote UGDH activity by maintaining the active site cysteine in a reduced state, and by recycling NAD+ from NADH. The level of oxygen in tissues and the number of mitochondria can have a role in this context?

4) Could authors discuss if the GSH can be considered a molecule useful in this context in preventing oxidative stress influencing HA synthesis?

Reviewer #2: In this manuscript, the authors have optimized an indirect detection system for activity of the enzyme UDP-glucose dehydrogenase (UGDH) in situ using primary fibroblast-like synoviocytes (FLS). The studies examined UGDH activity and hyaluronan (HA) production in response to known HA stimulatory factors PDGF, IL1�, and TGF�1 in cultured primary cells and in a rabbit model of joint injury.

Although the in situ assay for UGDH activity has potential qualitative value, there are a number of significant issues with the manuscript, some of the experiments, and the conclusions made by the authors. In fact, the overall purpose of the studies is difficult to ascertain clearly because the authors are inconsistent in their statements throughout the presentation of results and conclusions. In general, the report is more appropriate for a methods paper, as the biological data are too preliminary and speculative.

1. The assay is an indirect detection of reaction products generated by UGDH. As such, it cannot be used for quantitative comparisons of UGDH activity as presented by the authors. This is not a valid assay for quantification because it detects an amplified product at saturation. There are also no measurements that would validate the specificity of the assay. For example, other reports have used UDP-xylose, an endogenous inhibitor of UGDH, to demonstrate specificity, or a dose responsive UGDH -spiked standard, or blocked with excess UDP-glucuronate product, etc. The authors did not quantify the UDP-glucose, UDP-glucuronate, or UDP-xylose content of the cell culture, but those levels would be informative. There is also no examination of UGDH expression or HAS expression by western blot or qPCR, etc, to support the assertions made by the authors with respect to the relationship between UGDH activity and HA production.

2. Nuclear UGDH is not widely accepted as a biologically relevant phenomenon. In particular within these studies, where cells have been permeabilized by freezing and thawing, there is the significant possibility that integrity of organelles may also be partially breached. This is also potentially significant for sporadic UGDH loss from the cells. Imaging cannot be the sole method on which to base conclusions about location, activity, and relative expression of UGDH.

3. Conclusions about the redox state of the cells or tissues impacting UGDH activity in vivo based on peroxide, NEM, GR, or GSH in vitro or in situ are not supported by the data. These additives may impact the assay without having any physiological relevance, and the authors have not shown that UGDH is modified in the cells or tissues (e.g. by cysteine oxidation), nor even that these agents would modify purified UGDH protein.

4. Shedding of HA in conditioned media is highly sensitive to the confluence of the cells. Most of the HA assays were apparently done with cells at 90-100% confluence, where production would be reaching steady state so conclusions about its rate of synthesis and even its abundance are not valid at this point. Results in figure 1 should be considered carefully because it is unclear whether the treated cells were at the same confluence or viability when assayed following the respective treatments.

5. HAS and UGDH are both known to be post-translationally modified (both by phosphorylation and HAS also by N-acetylglucosaminylation), which impacts their activity. The authors have not measured or discussed this. In fact, UDP-glucuronate is not generally considered “rate limiting” for HA synthesis any longer due to this fact, and it has been reported that in some cells and conditions, UDP-N-acetylglucosamine is rate limiting for HA.

6. The authors appear to be extrapolating results from use of bacterial diaphorase and purified GR to suggest these enzymes affect UGDH activity in vivo. There is no evidence for this in any of their studies.

7. An excessive amount of results are inappropriately discussed in the introduction and the results narrative reads more appropriately for a methods paper or a review article, rather than primary literature reporting novel biological findings.

8. The numbering scheme in the multi-panel figures is incredibly confusing.

9. Several of the figures (e.g. 2, 3, 4, 5) could be placed in supplemental, as they do not contribute more than control data.

6. PLOS authors have the option to publish the peer review history of their article (what does this mean?). If published, this will include your full peer review and any attached files.

Reviewer #1: **Yes: **alberto passi

Reviewer #2: No

---

## [Author Response · Author response to Decision Letter 0]

22 Jul 2022

Dear Editor Gualillo, July 21, 2022

We would like to thank you for taking the time to review our paper, to find expert reviewers and allow us to respond to reviews. For your points 1 to 4: 

(1) the title page format was corrected

(2) requested text was added to Competing Interests, Ortho RTi was modified to Chitogenx, the new name of the company (changed effective around Aug. 1, 2022):

“I have read the journal's policy and the authors of this manuscript have the following competing interests: CDH, Scientific Advisory Board and a shareholder of Chitogenx Inc. (formerly Ortho RTi); RC, CM, RS, AC, DF, RM, HEG, SA and MP have declared that no competing interests exist. This does not alter our adherence to PLOS ONE policies on sharing data and materials.”

(3) Original uncropped and unadjusted images of our TLC plates have been uploaded as Supporting Information files that include a Word document with the lane contents. 

(4) as corresponding author (CDH), I have 3 affiliations related to this manuscript as the study was initiated at Polytechnique Montreal and completed at George Mason University. My current institute is George Mason University and should be the “chosen institute.” Please advise if something needs to be changed. 

(5) we added uncropped, original images of our TLC plates along with lane contents as supplemental data.

We hope that you and the reviewers will find our responses adequate and revised manuscript now acceptable for publication. 

5. Review Comments to the Author

Please use the space provided to explain your answers to the questions above. You may also include additional comments for the author, including concerns about dual publication, research ethics, or publication ethics. (Please upload your review as an attachment if it exceeds 20,000 characters) -- please note that we have uploaded an attachment "Response to Reviewers" with blue text answers where changes to the manuscript are underlined. Here is the same text below but missing underlines and modified figures. 

Reviewer #1: Hyaluronic acid is a polymer of enormous biological interest, and its concentration plays a decisive role in tissues both chemically and physiologically. This manuscript presents a histochemical method to measure the activity of the enzyme UGDH that synthesizes an HA precursor (UDPGlcUA). In the course of optimization, however, the authors were able to add an important piece to better understand the regulation of this enzyme in cellular and tissue models. The presented study aims to test whether stimulation of HA synthesis in cells involves a key enzyme for the synthesis of the HA precursor (UDPglucuronic) UGDH. UGDH has unique characteristics, for example, it is the only enzyme that can perform double oxidation on the same carbon, a unique feature in mammals.

The data do not support the hypothesis that UGDH is a rate-liming enzyme for HA synthesis under catabolic inflammatory conditions that can oxidize and inactivate the UGDH active site cysteine. In this way, the work better defines the differences between stimulation by PDGF and by cytokines, which act on the UGDH enzyme in completely different ways.

PDGF stimulates HA production and UGDH activity in primary human FLS whereas inflammatory cytokines stimulated dramatic HA production but without significant increases in UGDH enzyme activity.

The data are obtained by using diaphorase and glutathione reductase (GR) and showed that diaphorase or GR was necessary and sufficient to catalyze NADH reduction of formazan (NBT) to the level of colorimetric detection in a one-hour UGDH staining reaction.

The data are intriguing and the text well written, the text is clear and the figures well done. The literature is properly described.

Response: Thank you Dr. Passi for your supportive comments and appreciation of our work. 

There are few concerns that can be considered:

1) in line 563 the authors report “This study is the first, to our knowledge, to present a method for measuring in situ UGDH activity in cultured monolayer cells” but really there are a couple of papers addressing the question, in particular the positivity of the UGDH at the edge of cell cultures:

Castellani AA, De Luca G, Rindi S, Salvini R, Tira ME. Regulatory mechanisms of UDP-glucuronic acid biosynthesis in cultured human skin fibroblasts. Ital J Biochem. 1986 Sep-Oct;35(5):296-303. PMID: 3804697.

Rizzotti M, Cambiaghi D, Gandolfi F, Rindi S, Salvini R, De Luca G. The effect of extracellular matrix modifications on UDP-glucose dehydrogenase activity in cultured human skin fibroblasts. Basic Appl Histochem. 1986;30(1):85-92. PMID: 3718422

Nevertheless these papers are old and published in Journals not readily available and the techniques here reported are clearly updated.

Response: We read the suggested papers by Castellani et al (1986) and Rizzotti et al (1986) and in agreement with your comments, did not find any methods descriptions in these papers. The Castellani paper describes UGDH enzymatic assays on cell lysates and is not directly comparable to our work. The Rizzotti paper has figures showing histostained human skin fibroblasts, and staining methods referred to a book chapter (Aureli G et al., Basic and Applied Histochemistry 24, 1981). In this work, Aureli et al (1981) histostaining methods also cited previous work (Aureli G et al, Rivista di istochimica, normale e patologica. 15, 9-25, 1969; De Luca.et al., Basic and Appl. Histochem. 25, 67-69, 1981). The article by Aureli et al. (1969) stated that the method of Balogh and Cohen (1961) was used to stain cartilage cryosections without further detail.

De Luca et al. (1981) gives the following methods description: “…fibroblasts were allowed to grow on microscope slides in glass Leighton tubes with a surface area of 6 cm2, containing 2 mL of the previous medium. After 48 and 72 hours, standard medium was removed and replaced with fresh medium with or without 0.1 mM UDP-xylose and the cells were incubated 30 min at 20°C. After washing the cell layer for 5 min. with 0.1 M phosphate buffer, pH 8.2, the histoenzymological determination of UDPG-dehydrogenase was carried out for 30 min. in a reaction mixture prepared as described by Baloch at Cohen (1961)….It is still impossible to say whether UDP-xylose entered the fibroblasts and thus played a direct role…”

Although not explicitly stated, we infer from the text of De Luca et al. (1981) that live cells were incubated with the UGDH staining solution, which unfortunately could produce a false-positive stain due to NBT uptake by live cells and subsequent reduction by mitochondrial NADH reductases. Negative controls (staining with NBT and NAD+, no UDP-glucose) were missing. Furthermore, to our knowledge, there are no nucleotide-sugar transporters in the plasma membrane that can deliver UDP-glucose across the plasma membrane barrier to the cytosol of live cells in order to react with cytosolic UGDH enzyme.

To respond to this concern we have modified the results text accordingly (for new text -- please see the word document "Response to Reviewers" or track changes in our revised paper): 

Results, p20: "We were aware that all of the UGDH enzyme staining methods prior to our study used unfixed freeze-thawed tissues that we wanted to adapt for monolayer cultures. … De Luca et al. (26), Aureli et al. (27), and Rizzotti et al. (28) previously showed histostained monolayer chondrocytes and skin fibroblasts after incubating these cultured cells with the UGDH staining solution of Balogh et al. (15), however negative controls with NBT and no UDP-Glc substrate were missing, and the methods suggested that the staining solution was applied to live cells. We carefully considered all these questions while designing our staining procedure.”

2) the glucuronic acid is also a substrate for other glycosaminoglycans, and the authors did not mention it. Moreover it may be interesting to underline if the stimulation with PDGF ot TGFbeta and IL-1 alters the other GAG synthesis. Basically, is there a competition for the substrate in these cells? More HA is coupled with minor synthesis of other GAGs in Golgi? In these cells the carriers carrying UDPsugars in Golgi have very low Km in order to preserve GAG synthesis?

Response: These are intriguing ideas. We agree that UDPsugar partitioning between cytosol and ER must be critically regulated for chondrocytes which synthesize high levels of proteoglycan and hyaluronic acid. Neufeld et al. (1) proposed that UDP-xylose feedback inhibition of UGDH serves to prevent accretion of UDP-xylose when aggrecan expression is diminished. We were unable to find any publications analyzing whether PDGF versus TGF-beta1/IL-1beta differentially influence proteoglycan expression by primary cultured FLS, whose main GAG output is hyaluronic acid, or specific analyses of UXS in FLS. We did find papers generally describing UDP-GlcA and UDP-xylose transporters in the ER and Golgi (2-4, including your own work) and a review paper by Zimmer et al. (5) stated that these transporters are gradient-driven (passive). We were unable to find any explanation in the literature as to how mammalian cells maintain UDP-GlcA import to the ER in the face of high consumption of UDP-GlcA by HAS enzymes, in cell types with different levels of proteoglycan deposition. It is possible that the perinuclear UGDH staining activity that we detect in some FLS could be a clue as to how alterations in subcellular UGDH activity could help direct UDP-GlcA to the ER vs cytosol. We acknowledge that UDPsugar transporter regulation is outside the scope of our paper. These comments however along with some comments by Reviewer #2 led us to modify the discussion.

1. Neufeld, E. F.; Hall, C. W. Inhibition of UDP-D-Glucose Dehydrogenase by UDP-D-Xylose. A Possible Regulatory Mechanism/. Biochemical and Biophysical Research Communications 1961, 19 (4), 456–461. ISSN: 006-291X

2. Caon, I.; Parnigoni, A.; Viola, M.; Karousou, E.; Passi, A.; Vigetti, D. Cell Energy Metabolism and Hyaluronan Synthesis. J Histochem Cytochem. 2021, 69 (1), 35–47. https://doi.org/10.1369/0022155420929772.

3. Ashikov, A.; Routier, F.; Fuhlrott, J.; Helmus, Y.; Wild, M.; Gerardy-Schahn, R.; Bakker, H. The Human Solute Carrier Gene SLC35B4 Encodes a Bifunctional Nucleotide Sugar Transporter with Specificity for UDP-Xylose and UDP-N-Acetylglucosamine*. Journal of Biological Chemistry 2005, 280 (29), 27230–27235. https://doi.org/10.1074/jbc.M504783200.

4. Bakker, H.; Oka, T.; Ashikov, A.; Yadav, A.; Berger, M.; Rana, N. A.; Bai, X.; Jigami, Y.; Haltiwanger, R. S.; Esko, J. D.; Gerardy-Schahn, R. Functional UDP-Xylose Transport across the Endoplasmic Reticulum/Golgi Membrane in a Chinese Hamster Ovary Cell Mutant Defective in UDP-Xylose Synthase. 2009, 284 (4), 10.

5. Zimmer, B. M.; Barycki, J. J.; Simpson, M. A. Integration of Sugar Metabolism and Proteoglycan Synthesis by UDP-Glucose Dehydrogenase. J Histochem Cytochem. 2021, 69 (1), 13–23. https://doi.org/10.1369/0022155420947500.

Discussion page 29 (all new text and 3 new references):

“Our study has certain limitations in that we did not quantify UDP-Glc, UDP-GlcA, or UDP-xylose content in cultured FLS. It is admittedly challenging to measure the levels of these metabolites in biological systems. Others have reported that permeabilized microsomal membranes rapidly lose UDP-sugars (44). If freeze-thawed FLS monolayers led to UDP-xylose leakage into the cytosol, these hypothetical metabolite levels were insufficient to prevent UGDH activity staining in PDGF- and serum-stimulated FLS. We also cannot rule out the possibility that IL-1beta & TGFbeta1 selectively upregulated UDP-xylose synthase (UXS) activity and passive transport of UDP-xylose from the endoplasmic reticulum to the cytosol (45). However in chondrocytes (at least) UXS is subject to end product inhibition by UDP-xylose (46).” 

44. Bossuyt X, Blanckaert N. Carrier-mediated transport of intact UDP-glucuronic acid into the lumen of endoplasmic-reticulum-derived vesicles from rat liver. Biochem J. 1994;302(Pt 1):261–9. 

45. Zimmer BM, Barycki JJ, Simpson MA. Integration of Sugar Metabolism and Proteoglycan Synthesis by UDP-glucose Dehydrogenase. J Histochem Cytochem. 2021;69(1):13–23. 

46. John KV, Schwartz NB, Ankel H. UDP-glucuronate carboxy-lyase in cultured chondrocytes. Journal of Biological Chemistry. 1977;252(19):6707–10.

3) This study proposes new data and highlights that UGDH activity is controlled by a redox switch, where intracellular peroxide inactivates, and high glutathione and diaphorase promote UGDH activity by maintaining the active site cysteine in a reduced state, and by recycling NAD+ from NADH. The level of oxygen in tissues and the number of mitochondria can have a role in this context?

Response: Thank you for your comments. Synovial membranes are highly vascularized and are not typically considered to be under hypoxic stress, compared to cartilage, an avascular tissue. In our paper, all FLS cell cultures were carried out under normoxic conditions. Based on the work of others, we believe that stimulation of ROS and activation of mitochondrial superoxide dismutase is more likely to have a role than the number of mitochondria per se, in regulating endogenous peroxide levels in response to inflammation. We have added new text to the discussion, also in response to a concern raised by Reviewer 2 farther below (new text is shown in track changes of our revised paper):

Discussion, page 29: "These collective results can be reconciled as follows: IL-1β is known to stimulate FLS to produce reactive oxygen species (ROS) (47), which are converted to peroxide in the presence of mitochondrial superoxide dismutase (SOD2) (40). Peroxide is thus a common intracellular mediator of catabolic inflammation (48). Moreover, peroxide is an endogenous thiol-reactive compound that like NEM (Fig. 3), and other thiol-reactive chemicals (49), could inactivate UGDH by modifying active site Cys 276. It is noteworthy that UDP-xylose inhibits UGDH by masking the active site Cys 276 (50)."

References

44. Kamata H, Honda S-i, Maeda S, Chang L, Hirata H, Karin M. Reactive Oxygen Species Promote TNFα-Induced Death and Sustained JNK Activation by Inhibiting MAP Kinase Phosphatases. Cell. 2005 Mar 11;120(5):649–61. 

45. Campbell RE, Sala RF, Rijn I van de, Tanner ME. Properties and Kinetic Analysis of UDP-glucose Dehydrogenase from Group A Streptococci: irreversible inhibition by UDP-chloroacetol. Journal of Biological Chemistry. 1997;272(6):3416–22. 

50. Kadirvelraj R, Sennett NC, Polizzi SJ, Weitzel S, Wood ZA. Role of Packing Defects in the Evolution of Allostery and Induced Fit in Human UDP-Glucose Dehydrogenase. Biochemistry. 2011 Jun 28;50(25):5780–9.

4) Could authors discuss if the GSH can be considered a molecule useful in this context in preventing oxidative stress influencing HA synthesis?

Response: We are uncertain whether oxidative stress dynamics are sufficient to regulate HA synthesis in cultured FLS. Because the principle aim of our study was to analyze whether cytokines that enhance HA production stimulate UGDH activity, we did not test whether modulation of GSH levels influences HA production. In the cancer field, chronic anti-oxidants including N-acetyl cysteine (NAC) were tested for their ability to limit HA production which is tied to cancer progression. However at least 2 studies showed that this approach backfired by promoting metastasis. These paradoxical findings have yet to be explained, however our data now open the possibility that NAC treatment may have helped maintain UGDH activity which according to Arnold et al. 2020, promotes EMT. We believe that our data showing strong A549 UGDH staining activity is an important finding that gives our work broader impact and interest to readers of PLoS ONE:

Discussion, page 31: “UGDH activity was implicated to-date in lung adenocarcinoma, breast cancer and glioblastoma (29–31,56). Our data additionally implicate diaphorase, GR, peroxide, and GSH in this process. Our findings provide potential explanations for the paradoxical metastasis-promoting effects of anti-oxidants (57,58).”

References: 

30. Arnold JM, Gu F, Ambati CR, Rasaily U, Ramirez-Pena E, Joseph R, et al. UDP-Glucose 6-Dehydrogenase regulates hyaluronic acid production and promotes breast cancer progression. Oncogene. 2020;27.

57. Sayin VI, Ibrahim MX, Larsson E, Nilsson JA, Lindahl P, Bergo MO. Antioxidants Accelerate Lung Cancer Progression in Mice. Sci Transl Med [Internet]. 2014 Jan 29 [cited 2021 Dec 15];6(221). Available from: https://www.science.org/doi/10.1126/scitranslmed.3007653

58. Le Gal K, Ibrahim MX, Wiel C, Sayin VI, Akula MK, Karlsson C, et al. Antioxidants can increase melanoma metastasis in mice. Sci Transl Med. 2015 Oct 7;7(308):308re8.

Reviewer #2: In this manuscript, the authors have optimized an indirect detection system for activity of the enzyme UDP-glucose dehydrogenase (UGDH) in situ using primary fibroblast-like synoviocytes (FLS). The studies examined UGDH activity and hyaluronan (HA) production in response to known HA stimulatory factors PDGF, IL1beta, and TGFbeta1 in cultured primary cells and in a rabbit model of joint injury.

Although the in situ assay for UGDH activity has potential qualitative value, there are a number of significant issues with the manuscript, some of the experiments, and the conclusions made by the authors. In fact, the overall purpose of the studies is difficult to ascertain clearly because the authors are inconsistent in their statements throughout the presentation of results and conclusions. In general, the report is more appropriate for a methods paper, as the biological data are too preliminary and speculative. 

Response: We thank the reviewer for taking the time to read our paper carefully and make thoughtful comments that have helped us to improve our work. We are uncertain which statements you found to be inconsistent. With all due respect, we beg to differ that our biological data are preliminary and speculative. Our methodology builds on the work of Balogh et al (1961), Mehdizadeh et al (1991), and Pitsillides et al (1992a, 1992b). We carefully optimized and detailed our findings in order to vet rigor of prior research, and to allow our peers to reproduce our results or use our methods with other cell types. We present results based on studies of primary FLS cells from 6 different biological donors, 6 synovial specimens from 3 New Zealand White rabbits submitted to unilateral cartilage repair of a chronic defect under ethics-approved protocols, and our data are backed up by enzyme histostaining experiments in cell lines, cell-free enzymatic assays, chemical characterization of the mechanisms involved in generating the formazan product, and a logical integration and fit of our data with a large body of literature (66 citations). We have ensured that all experiments have biological and technical replicates and appropriate controls. In our response to these general criticisms, we wish to draw your attention to the supportive comments of Reviewer 1, who agreed to disclose his identity in a transparent review process. Dr. Passi is an internationally-recognized expert in the field of hyaluronic acid synthesis and UGDH regulation, and his comments show that he finds merit in our findings. Please find below our detailed responses to each of the concerns.

1. The assay is an indirect detection of reaction products generated by UGDH. As such, it cannot be used for quantitative comparisons of UGDH activity as presented by the authors. 

Response: We agree that the UGDH histostain is an indirect detection method and that we are measuring reaction products (i.e., NADH generation by UGDH leading to NBT reduction). Biochemical UGDH activity assays are likewise most often based on indirect detection of NADH absorbance by spectrophotometry whereas in our assay, we use microscopic digital camera pixel intensity to quantify the NBT reaction product. Our statistical analyses show that the probability of obtaining a result as extreme or more extreme as the result we observed in FLS stimulated by various cytokines, is less than the cut-off of p=0.05, which leads us to reject the null hypothesis that cytokines have no effect on UGDH activity, because serum and PDGF induced a significantly higher staining intensity than serum-free cultures for N=5 biological replicates. Our data also allow us to reject the null hypothesis that peroxide, NEM, and UDP-xylose have no effect on UGDH activity. To specifically respond to these comments and emphasize the validity of our approach, we modified the abstract and the discussion:

Abstract: "We used enzyme histostaining and quantitative image analysis to test whether cytokines that stimulate HA synthesis upregulate UGDH activity."

Abstract: "Primary synovial fibroblasts and transformed A549 fibroblasts showed constitutive diaphorase/GR staining activity that varied according to supplied NADH levels, with relatively stronger UGDH and diaphorase activity in A549 cells." 

Discussion (page 30, second sentence is new text). "Our study provides novel evidence that UGDH enzyme histostaining is 2-factor mechanism. In our cell-free assay, we observed that 1 mM NADH alone cannot spontaneously reduce NBT effectively after 1 hour, and requires diaphorase or GR to accelerate the reduction. Differences in UGDH activity staining were not due to altered diaphorase/GR activity because freeze-thawed cells showed strong constitutively active diaphorase/GR histostaining, irrespective of the presence of peroxide, NEM, or UDP-xylose." 

This is not a valid assay for quantification because it detects an amplified product at saturation.

Response: We apologize for lack of clarity in our paper, in that we have pre-emptively addressed the concern of signal saturation, but the text needs to be further clarified. Firstly, we stained cells with NADH at saturating (1 mM) and non-saturating (0.1 mM) concentrations. The data shown in our original submitted manuscript suggested that a 1 hour UGDH staining reaction, which includes 1 mM NAD+, does not reach saturation because the staining intensity levels are closer to those obtained for 0.1 mM NADH and not 1 mM NADH, as shown in Figure 3 (formerly Figure 4, panels re-organized for clarity as requested farther below).

Figure 3. UGDH enzyme histostaining was inhibited when NEM was included in the enzyme staining solution (“in stain”) whereas diaphorase/GR staining was only partly inhibited when live cells were pre-treated with 1 mM NEM for 2 hours before enzyme staining (“added to live cells”). (A) Enzyme activity was measured by % pixel staining intensity of monolayer cells (median, 25% quartile range (box), min-max, distinct cultures with 2 female human donors and 1 male donor FLS, n=4-7). Panels (B-E) show representative enzyme staining results for FLS from one female donor for (B1-B3) NAD+ alone (negative control), (C1-C3) UDP-Glc and NAD+ (UGDH stain) (D1-D3) 0.1 mM NADH and (E1-E3) 1 mM NADH (diaphorase/GR stain). Columns show (B1-E1) no NEM, (B2-E2) with NEM added “in stain”, or (B3-E3) NEM “added to live cells”. In panel A, significant differences are noted with letters and symbols (*, ***). Scale bars: 50 µm.

Secondly, we present data showing that longer enzyme staining incubation times can lead to stronger UGDH staining activity, even in the presence of low levels of peroxide, at peroxide levels previously shown to have a reversible inhibitory effect on enzymes with an active site cysteine (Figure 6, formerly Figure 7).

Figure 6. UGDH enzyme activity in synovial lining cells (A, I, M) was inhibited by NEM (D) and by peroxide in a dose-dependent manner (B, C, J, N, K, O) but “diaphorase” was insensitive to thiol modifying inhibitors (F, G, H, L). UGDH and “diaphorase” enzyme histostaining was carried out per standard conditions, in the presence of peroxide or NEM, or with NAD+ only (±1 mM GSH) as indicated, in synovial cryosections from a New Zealand White male rabbit knee 1 week post-microdrilling, with a 60 or 75 minute incubation at 37°C. UGDH enzyme staining was completely suppressed by NEM (D) and more suppressed by 1 mM peroxide (C, K, O) than 50 µM peroxide (B, J, N). Scale bars are 20 µm or 100 µm, as indicated

To specifically respond to this concern, the text was revised accordingly:

Results page 25: (please see track changes in our revised paper) Header: “NEM is a potent inhibitor of UGDH activity in freeze-thawed and live FLS”;

“……FLS were stained in parallel for “diaphorase” by incubating freeze-thawed monolayers with NBT and low NADH (0.1 mM NADH) or high NADH (1 mM, which simulates 100% conversion of 1 mM NAD+ by UGDH). These “diaphorase” staining conditions produced either a pale (0.1 mM NADH) or an intense dark blue (1 mM NADH) granular and insoluble diformazan in both in the cytosol and nucleus of FLS cells (Fig. 3D & 3E, respectively). FLS showed on average 17% or 53% diaphorase staining with low and high NADH, respectively, and 7% UGDH enzyme staining activity (Fig. 3A). These results suggested that the UGDH staining reaction had not reached saturation. FLS showed on average 17% or 53% staining with low and high diaphorase, respectively, and 7% UGDH enzyme staining activity (Fig. 3A). These results suggested that the UGDH staining reaction had not reached saturation.” 

There are also no measurements that would validate the specificity of the assay. For example, other reports have used UDP-xylose, an endogenous inhibitor of UGDH, to demonstrate specificity, or a dose responsive UGDH -spiked standard, or blocked with excess UDP-glucuronate product, etc. 

Response: We acknowledge that UDP-xylose is an excellent control to include in our study. To address this concern, we carried out new experiments with UDP-D-xylose added to the staining reaction. The results show that 0.2 mM UDP-xylose inhibits the staining reaction (please see new Figure S5). 

Figure S5. UDP-xylose and peroxide inhibited UGDH but not diaphorase staining activity in cultured monolayer FLS. FLS stained for UGDH (A: bright field; B: phase contrast) or UGDH with UDP-xylose (C: bright field; D: phase contrast), diaphorase (E), or diaphorase with UDP-xylose (F). In panel G, quantitative histomorphometry was used to measure the relative staining intensity in the absence or presence of UDP-xylose or peroxide. Symbols: *p<0.01 vs UGDH; ** p<0.0001 vs UGDH. Scale bars: 50 µm.

The text was modified accordingly (for new text please see track changes in the revised paper):

Introduction, Page 6: “Following these insights, we systematically analyzed whether N-ethyl maleimide (NEM, a reported inhibitor of GR and diaphorase), peroxide (an inflammatory mediator), or UDP-xylose (an allosteric inhibitor of UGDH (19)), can inhibit these enzyme staining reactions.” 

Methods (page 12): For negative controls, UDP-glucose was substituted with ddH2O, or UDP-xylose (Chemilys, GA) was added at 0.2 mM (24) to the UGDH staining solution. 

Results page 26: “Finally, addition of 0.2 mM UDP-xylose, a well-known allosteric inhibitor of UGDH (19), or 0.05 mM peroxide, to the staining solution strongly suppressed UGDH but not diaphorase staining (0.02 or 0.15 pixels/cell, respectively, vs 1.48 pixels/cell, p<0.01, Fig. S5).”

Discussion page 29: “It is noteworthy that UDP-xylose inhibits UGDH by masking the active site Cys 276 (50).”

References (19 & 50 are new references)

19. Neufeld EF, Hall CW. Inhibition of UDP-D-glucose dehydrogenase by UDP-D-xylose. A possible regulatory mechanism/. Biochemical and Biophysical Research Communications. 1961;19(4):456–61. 

24. Mehdizadeh S, Bitensky L, Chayen J. The assay of uridine diphosphoglucose dehydrogenase activity: Discrimination from xanthine dehydrogenase activity. Cell Biochemistry and Function. 1991 Apr 1;9(2):103–10.

50. Kadirvelraj R, Custer GS, Keul ND, Sennett NC, Sidlo AM, Walsh RM, et al. Hysteresis in Human UDP-Glucose Dehydrogenase Is Due to a Restrained Hexameric Structure That Favors Feedback Inhibition. Biochemistry. 2014 Dec 30;53(51):8043–51. 

The authors did not quantify the UDP-glucose, UDP-glucuronate, or UDP-xylose content of the cell culture, but those levels would be informative. There is also no examination of UGDH expression or HAS expression by western blot or qPCR, etc, to support the assertions made by the authors with respect to the relationship between UGDH activity and HA production.

Response: This is correct, we did not quantify UDP-glucose, UDP-glucuronate, or UDP-xylose content of the cell culture, but we did analyze UGDH protein expression by immunohistology. Recklies et al (citation 11) showed that HAS gene expression fails to correlate with HA release in cytokine-stimulated FLS from OA and RA patients, and her negative findings partly motivated this study.

The lack of these measures (UDPsugars) does not alter our novel findings that NEM, peroxide, and UDP-xylose inhibit UGDH staining activity, and that PDGF enhances both UGDH activity and HA synthesis, and that TGF-beta1+IL-1beta does not stimulate the same 27-fold increase in UGDH activity as the observed 27-fold increase in HA release. 

To address these concerns we added these limitations to our discussion: 

Discussion page 29 (all new text and 3 new references): 

“Our study has certain limitations in that we did not quantify UDP-Glc, UDP-GlcA, or UDP-xylose content in the cell cultures. It is admittedly challenging to measure the levels of these metabolites in biological systems. Others have reported that permeabilized microsomal membranes rapidly lose UDP-sugars (42). If freeze-thawed FLS monolayers led to UDP-xylose leakage into the cytosol, these hypothetical metabolite levels were insufficient to prevent UGDH activity staining in PDGF- and serum-stimulated FLS. We also cannot rule out the possibility that IL-1β & TGF- β1 selectively upregulated UDP-xylose synthase (UXS) activity and passive transport of UDP-xylose from the endoplasmic reticulum to the cytosol (45). However in chondrocytes (at least) UXS is subject to end product inhibition by UDP-xylose (46). We also did not measure UGDH gene expression by Western blot or RT-PCR, but immunostaining showed relatively homogeneous cytosolic UGDH expression and occasional nuclear staining in FLS monolayers from 5 different human donors under all cytokine conditions.”

References

42. Bossuyt X, Blanckaert N. Carrier-mediated transport of intact UDP-glucuronic acid into the lumen of endoplasmic-reticulum-derived vesicles from rat liver. Biochem J. 1994 Aug 15;302(Pt 1):261–9.

45. Zimmer BM, Barycki JJ, Simpson MA. Integration of Sugar Metabolism and Proteoglycan Synthesis by UDP-glucose Dehydrogenase. J Histochem Cytochem. 2021 Jan;69(1):13–23.

46. John KV, Schwartz NB, Ankel H. UDP-glucuronate carboxy-lyase in cultured chondrocytes. Journal of Biological Chemistry. 1977 Oct;252(19):6707–10.

2. Nuclear UGDH is not widely accepted as a biologically relevant phenomenon. 

Response: However controversial, UGDH nuclear localization data are published in peer-reviewed reputable journals, and therefore important to recognize. 

Reference section (citations 29 and 30 in our original manuscript):

32. Arnold JM, Gu F, Ambati CR, Rasaily U, Ramirez-Pena E, Joseph R, et al. UDP-Glucose 6-Dehydrogenase regulates hyaluronic acid production and promotes breast cancer progression. Oncogene. 2020;27. 

33. Hagiuda D, Nagashio R, Ichinoe M, Tsuchiya B, Igawa S, Naoki K, et al. Clinicopathological and prognostic significance of nuclear UGDH localization in lung adenocarcinoma. Biomedical Research. 2019;40(1):17–27. 

Although, from these comments we recognized that our descriptions of our protein expression data could be improved. We have modified the Results section and added a new supplemental figure.

 Results page 19 (please see new figure S2): “Anti-UGDH immunostaining produced a cytosolic stain in all 4 conditions for FLS from 5 different donors (Fig. 1A1-D1). UGDH was also detected in the nucleus in selected serum- or cytokine-stimulated cells (Fig. S2).”

Fig. S2. UGDH protein detected by immunostaining localized mainly to the cytosol (white arrows) with a cytosolic and detectable nuclear stain (open arrows) in selected cells. Cultured FLS (P4) extracted from a synovial biopsy from 23-year old male donor were cultured under different cytokine conditions as indicated, fixed in acetone, and immunostained for UGDH with red substrate detection which is fluorescent. Panels show phase contrast (A-D) and matching epifluorescent images of UGDH immunostain (E-H) and DAPI-counterstained fluorescent nuclei (I-L). Scale bars: 20 µm. 

In particular within these studies, where cells have been permeabilized by freezing and thawing, there is the significant possibility that integrity of organelles may also be partially breached. This is also potentially significant for sporadic UGDH loss from the cells. Imaging cannot be the sole method on which to base conclusions about location, activity, and relative expression of UGDH.

Response: Enzyme histochemistry is both a historical, and modern method that can use PVA to retain enzymes in unfixed specimens (Ref. 30 below). The level of PVA in the staining solutions was optimized during our study and was a controlled variable in all our staining reactions. To address this concern, we added a citation to our Results section, and modified the conclusion: 

Results page 21: “With our method, we found that 8% PVA in Gly-Gly (pH 7.8) could be prepared and diluted to 4.2% or 5.3% w/v working concentration and prevent UGDH enzyme from diffusing out of the cells (30).”

(30) Feder, N. Polyvinyl alcohol as an embedding medium for lipid and enzyme histochemistry J. Histochem Cytochem. 1962, 10:341. DOI:10.1177/10.3.341.

Conclusions page 35: “We report a new method to detect in situ UGDH activity in cultured cells, through a 2-enzyme NBT-based histostaining reaction that requires both UGDH and endogenous diaphorase or GR for colorimetric detection. The staining method can be used to measure UGDH activity in cultured cells, and to screen specific factors for their ability to promote or inhibit UGDH activity.”

3. Conclusions about the redox state of the cells or tissues impacting UGDH activity in vivo based on peroxide, NEM, GR, or GSH in vitro or in situ are not supported by the data. These additives may impact the assay without having any physiological relevance, and the authors have not shown that UGDH is modified in the cells or tissues (e.g. by cysteine oxidation), nor even that these agents would modify purified UGDH protein.

Response: A large volume of literature shows that active site cysteines in many enzymes can be controlled by redox potential. Our data showing that the same levels of peroxide that modify other active site cysteine enzymes can also inhibit UGDH activity, which connects the dots between a large volume of literature describing the role of active site Cys 276 in UGDH biology, to redox dynamics in FLS. 

We show in Figure 3 (formerly Figure 4) that NEM specifically inhibits UGDH and not diaphorase enzyme histostaining activity at 0.1 mM. We show in Figure S3 that UDP-xylose specifically inhibits UGDH staining activity at 0.2 mM. We show in Figure 5 that peroxide specifically inhibits UGDH activity at 0.05 mM (see figures shown above, in response to concern #1). Others have crystallized active site cysteine enzymes after exposure to peroxide, as we mention in our discussion, and showed that this can lead to chemical modification of Cys and inactivation of the active site. As a breakthrough discovery, our work opens up new avenues of experimentation with purified UGDH protein. 

To specifically respond to this concern, we have added a new citation (Campbell et al, 1997) that details a number of thiol-reactive compounds that react with active site cysteine in UGDH leading to loss of UGDH enzymatic activity, and modified the discussion text accordingly:

Discussion, pages 29 and 30: “…These collective results can be reconciled as follows: IL-1beta is known to stimulate FLS to produce reactive oxygen species (ROS) (43), which are converted to peroxide in the presence of mitochondrial superoxide dismutase (SOD2) (38). Peroxide is thus a common intracellular mediator of catabolic inflammation (44). Peroxide is an endogenous thiol-reactive compound that like NEM (Fig. 3) and other thiol-reactive chemicals (45), could inactivate UGDH by reaction with active site Cys 276. It is noteworthy that UDP-xylose inhibits UGDH by masking the active site Cys 276 (50).”

“…Our data showed that UGDH activity, which depends on a catalytic Cys (1,2,53), is suppressed by 50 µM to 1 mM peroxide (i.e., at levels slightly above the Km 34.4 µM of human UGDH for UDP-Glc (54), Fig. 6).”

References

44. Kamata H, Honda S ichi, Maeda S, Chang L, Hirata H, Karin M. Reactive Oxygen Species Promote TNFα-Induced Death and Sustained JNK Activation by Inhibiting MAP Kinase Phosphatases. Cell. 2005 Mar 11;120(5):649–61. 

45. Campbell RE, Sala RF, Rijn I van de, Tanner ME. Properties and Kinetic Analysis of UDP-glucose Dehydrogenase from Group A Streptococci: irreversible inhibition by UDP-chloroacetol *. J Biol Chem. 1997 Feb 7;272(6):3416–22. 

54. Hyde AS, Thelen AM, Barycki JJ, Simpson MA. UDP-glucose Dehydrogenase Activity and Optimal Downstream Cellular Function Require Dynamic Reorganization at the Dimer-Dimer Subunit Interfaces. J. Biol Chem. 2013;288:35049–57.

4. Shedding of HA in conditioned media is highly sensitive to the confluence of the cells. Most of the HA assays were apparently done with cells at 90-100% confluence, where production would be reaching steady state so conclusions about its rate of synthesis and even its abundance are not valid at this point. Results in figure 1 should be considered carefully because it is unclear whether the treated cells were at the same confluence or viability when assayed following the respective treatments.

Response: The methods we used to measure HA production rates in FLS were based on those used by Recklies et al. 2001 (cited 96 times, Web of Science) who analyzed HA production rate in cultured FLS extracted from OA or RA synovium, using 4-day conditioned medium. We collected the 24-hour conditioned medium from our FLS cultures that were seeded for all conditions at the same cell density and furthermore normalized HA levels to cell number using guanidine-extracted monolayer cells and the Hoechst DNA assay against a standard curve of calf thymus DNA. We obtained comparable results to those of Recklies et al. 2001 (if you correct for number of days in culture). Because we did not see a correlation between UGDH staining intensity and HA production, we do not believe that repeating the cultures at different levels of confluency would change our results. To address this concern we clarified that our data were comparable to Recklies et al: 

Results page 18: “These data verified that primary FLS from donors free of chronic joint disease expressed UGDH and responded to cytokines with different HA production rates that were comparable to HA synthetic rates previously measured in cytokine-stimulated OA or RA FLS monolayers (11).”

References:

11. Recklies AD, White C, Melching L, Roughley PJ. Differential regulation and expression of hyaluronan synthases in human articular chondrocytes, synovial cells and osteosarcoma cells. Biochem J. 2001 Feb 15;354(Pt 1):17–24.

5. HAS and UGDH are both known to be post-translationally modified (both by phosphorylation and HAS also by N-acetylglucosaminylation), which impacts their activity. The authors have not measured or discussed this. In fact, UDP-glucuronate is not generally considered “rate limiting” for HA synthesis any longer due to this fact, and it has been reported that in some cells and conditions, UDP-N-acetylglucosamine is rate limiting for HA.

Response: Thankyou for your comments. We were unable to find any publications that measured the levels of the UDP-N-acetylglucosamine metabolite in primary FLS (without or with cytokine stimulation) and are unsure of how to address this comment. To address the comment on post-translational modifications, we have added text to the discussion and a new citation:

Discussion page 27: “Our results do not support the hypothesis that UGDH activity is rate-limiting for HA synthesis because IL-1beta & TGFbeta1 stimulated 27-fold higher HA production, potentially through increased hyaluronic acid synthase 2 (HAS2) expression (11,40), or increased HAS activity via post-translational modification (41), without proportional increases in UGDH activity.”

Results

41. Kasai K, Kuroda Y, Takabuchi Y, Nitta A, Kobayashi T, Nozaka H, et al. Phosphorylation of Thr328 in hyaluronan synthase 2 is essential for hyaluronan synthesis. Biochemical and Biophysical Research Communications. 2020 Dec;533(4):732–8. 

6. The authors appear to be extrapolating results from use of bacterial diaphorase and purified GR to suggest these enzymes affect UGDH activity in vivo. There is no evidence for this in any of their studies.

Response: We would like to emphasize that data in Figure 2 (formerly Fig. 3A) show that a diaphorase, or GR, is necessary and sufficient for NADH (supplied by UGDH) to reduce NBT in a 1-hour reaction time at 37°C. We also provide clear evidence with the histostaining data in Figures 3, 4, 5, 6 and S5 that cultured FLS express constitutively active endogenous diaphorase enzyme(s) that resist inhibition by thiol-modifying (NEM, peroxide) or masking (UDP-xylose) agents added to the staining solution. Given that UGDH activity is stimulated by NAD+, and that both diaphorase and GR were shown to drive the reduction of NBT by NADH which regenerates NAD+, collective data from our study support the mechanistic histostaining model that we propose. To address this comment, we have modified the header in Figure 7 (formerly Figure 8) by adding the term "hypothetical" to emphasize that our model, which Dr. Passi finds to be entirely plausible, is hypothetical, and added UDP-xylose to the figure as well. 

Figure 7. Hypothetical model representing 2-enzyme UGDH activity staining mechanisms in cultured cells and unfixed cryosections.

7. An excessive amount of results are inappropriately discussed in the introduction and the results narrative reads more appropriately for a methods paper or a review article, rather than primary literature reporting novel biological findings.

Response: We apologize for the lengthy introduction and have trimmed results from the text. 

8. The numbering scheme in the multi-panel figures is incredibly confusing.

Response: We apologize for the confusing layout. We have re-arranged the multi-panel figures and put some of the content in supplemental figures, as suggested in point 9.

9. Several of the figures (e.g. 2, 3, 4, 5) could be placed in supplemental, as they do not contribute more than control data.

Response: Please see response to point 8 above (a montage of the re-numbered figures can be seen in our file "Response to Reviewers"). 

6. PLOS authors have the option to publish the peer review history of their article (what does this mean?). If published, this will include your full peer review and any attached files.

Response: We agree to have our peer review history published.

Do you want your identity to be public for this peer review? For information about this choice, including consent withdrawal, please see our Privacy Policy.

Reviewer #1: Yes: alberto passi

Reviewer #2: No

---

## [Decision Letter · Decision Letter 1]

15 Aug 2022

PONE-D-22-03359R1UDP-glucose dehydrogenase (UGDH) activity is suppressed by peroxide and promoted by PDGF, diaphorase and glutathione reductase in fibroblast-like synoviocytesPLOS ONE

Dear Dr. Caroline D Hoemann,

Thank you for submitting your manuscript to PLOS ONE. After careful consideration, we feel that it has merit but does not fully meet PLOS ONE’s publication criteria as it currently stands. Therefore, we invite you to submit a revised version of the manuscript that addresses the points raised during the review process.

We look forward to receiving your revised manuscript.

Kind regards,

Abdelwahab Omri, Pharm B, Ph.D, Laurentian University  Canada

Academic Editor

PLOS ONE

Reviewers' comments:

Reviewer's Responses to Questions

**Comments to the Author**

1. If the authors have adequately addressed your comments raised in a previous round of review and you feel that this manuscript is now acceptable for publication, you may indicate that here to bypass the “Comments to the Author” section, enter your conflict of interest statement in the “Confidential to Editor” section, and submit your "Accept" recommendation.

Reviewer #1: All comments have been addressed

Reviewer #2: (No Response)

2. Is the manuscript technically sound, and do the data support the conclusions?

Reviewer #1: Yes

Reviewer #2: Partly

3. Has the statistical analysis been performed appropriately and rigorously? 

Reviewer #1: Yes

Reviewer #2: Yes

4. Have the authors made all data underlying the findings in their manuscript fully available?

Reviewer #1: Yes

Reviewer #2: Yes

5. Is the manuscript presented in an intelligible fashion and written in standard English?

Reviewer #1: Yes

Reviewer #2: Yes

6. Review Comments to the Author

Reviewer #1: the authors have greatly improved the text by clearly responding point by point to all the comments raised in the previous version of the manuscript

Reviewer #2: In this revised manuscript, the authors have strongly addressed the majority of the prior critiques. The manuscript reports and interprets intriguing biological phenomena related to arthritis and its associated pathologies utilizing a combination of human specimens and an elegant rabbit model of arthritic injury. A number of technical issues have been thoroughly addressed and discussed in a manner that will have significant value to researchers in a variety of related fields.

The key lingering concern is that some of the biological interpretations emphasized by the authors have not been explicitly validated by the authors. The point of concern is not that the speculation about peroxide oxidation of an active site cysteine, and its resolution by redox buffering proteins like glutathione reductase and diaphorase, is not feasible, valid, or supported by the data they have generated, but rather that some of the interpretations are subjective and there are alternatives that are not fully considered by the authors as indicated in the prior review. The title of the manuscript immediately prompts the expectation that the biological interpretation particularly surrounding redox potential and driving sources of redox buffering is the experimental focus of the paper. However, the interpretation actually relies on a selection of published results that have not been reproduced by the authors. Simply softening the title would mitigate this expectation and better represent the impact of the manuscript. This would be an adequate solution to the biological interpretation concerns.

The authors made the following observation in their response: “…we wish to draw your attention to the supportive comments of Reviewer 1, who agreed to disclose his identity in a transparent review process. Dr. Passi is an internationally-recognized expert in the field of hyaluronic acid synthesis and UGDH regulation, and his comments show that he finds merit in our findings.”

Though it does not influence this review or the determination of the manuscript’s significance, this statement offers an opportunity for a professional development comment. Three implications derive from this statement and how it is worded: 1) that revelation of reviewer identity indicates review transparency and reviewer competency; 2) that identity of the reviewer shows research merit; and 3) that failure to reveal identity indicates lack of competence or integrity - or even support - in critical evaluation of scientific publication. PLOS One is one of several progressive journals that offer the option for reviewers to reveal their identity. There are many reasons, both positive and negative, for reviewers to reveal or conceal identity. Many of the most qualified scientific reviewers will decline to review if their identity is compulsorily made public. Competency of the reviewer can be presumed by the quality of the review comments and the reviewer’s acceptance of the opportunity to review the response. It is easy to deflect a critical review by denigrating the character or academic record of an experienced reviewer if compelled to reveal identity, for example, rather than accepting the comments and responding to them in good faith. Moreover, journal editors vet the quality and expertise of scientific reviewers before inviting them to review a manuscript. The statement made by the authors can be interpreted as an accusation to the editor and the reviewer that there is a lack of expertise in the review since the reviewer is anonymous to the author. The author now has the choice to submit to a range of journals with policies that do not permit anonymity if they do not have faith in anonymous review.

7. PLOS authors have the option to publish the peer review history of their article (what does this mean?). If published, this will include your full peer review and any attached files.

Reviewer #1: **Yes: **alberto passi

Reviewer #2: No

---

## [Author Response · Author response to Decision Letter 1]

26 Aug 2022

Reviewers' comments:

Reviewer's Responses to Questions

Comments to the Author

1. If the authors have adequately addressed your comments raised in a previous round of review and you feel that this manuscript is now acceptable for publication, you may indicate that here to bypass the “Comments to the Author” section, enter your conflict of interest statement in the “Confidential to Editor” section, and submit your "Accept" recommendation.

Reviewer #1: All comments have been addressed

Reviewer #2: (No Response)

2. Is the manuscript technically sound, and do the data support the conclusions?

Reviewer #1: Yes

Reviewer #2: Partly

3. Has the statistical analysis been performed appropriately and rigorously? 

Reviewer #1: Yes

Reviewer #2: Yes

4. Have the authors made all data underlying the findings in their manuscript fully available?

Reviewer #1: Yes

Reviewer #2: Yes

5. Is the manuscript presented in an intelligible fashion and written in standard English?

Reviewer #1: Yes

Reviewer #2: Yes

6. Review Comments to the Author

Reviewer #1: the authors have greatly improved the text by clearly responding point by point to all the comments raised in the previous version of the manuscript

Reviewer #2: In this revised manuscript, the authors have strongly addressed the majority of the prior critiques. The manuscript reports and interprets intriguing biological phenomena related to arthritis and its associated pathologies utilizing a combination of human specimens and an elegant rabbit model of arthritic injury. A number of technical issues have been thoroughly addressed and discussed in a manner that will have significant value to researchers in a variety of related fields.

The key lingering concern is that some of the biological interpretations emphasized by the authors have not been explicitly validated by the authors. The point of concern is not that the speculation about peroxide oxidation of an active site cysteine, and its resolution by redox buffering proteins like glutathione reductase and diaphorase, is not feasible, valid, or supported by the data they have generated, but rather that some of the interpretations are subjective and there are alternatives that are not fully considered by the authors as indicated in the prior review. The title of the manuscript immediately prompts the expectation that the biological interpretation particularly surrounding redox potential and driving sources of redox buffering is the experimental focus of the paper. However, the interpretation actually relies on a selection of published results that have not been reproduced by the authors. Simply softening the title would mitigate this expectation and better represent the impact of the manuscript. This would be an adequate solution to the biological interpretation concerns.

Response: we thank the reviewer for these comments and have modified the title of our paper.

Original title:

UDP-glucose dehydrogenase (UGDH) activity is suppressed by peroxide and promoted by PDGF, diaphorase and glutathione reductase in fibroblast-like synoviocytes 

Modified title:

UDP-glucose dehydrogenase (UGDH) activity is suppressed by peroxide and promoted by PDGF in fibroblast-like synoviocytes: evidence of a redox control mechanism

The authors made the following observation in their response: “…we wish to draw your attention to the supportive comments of Reviewer 1, who agreed to disclose his identity in a transparent review process. Dr. Passi is an internationally-recognized expert in the field of hyaluronic acid synthesis and UGDH regulation, and his comments show that he finds merit in our findings.”

Though it does not influence this review or the determination of the manuscript’s significance, this statement offers an opportunity for a professional development comment. Three implications derive from this statement and how it is worded: 1) that revelation of reviewer identity indicates review transparency and reviewer competency; 2) that identity of the reviewer shows research merit; and 3) that failure to reveal identity indicates lack of competence or integrity - or even support - in critical evaluation of scientific publication. PLOS One is one of several progressive journals that offer the option for reviewers to reveal their identity. There are many reasons, both positive and negative, for reviewers to reveal or conceal identity. Many of the most qualified scientific reviewers will decline to review if their identity is compulsorily made public. Competency of the reviewer can be presumed by the quality of the review comments and the reviewer’s acceptance of the opportunity to review the response. It is easy to deflect a critical review by denigrating the character or academic record of an experienced reviewer if compelled to reveal identity, for example, rather than accepting the comments and responding to them in good faith. Moreover, journal editors vet the quality and expertise of scientific reviewers before inviting them to review a manuscript. The statement made by the authors can be interpreted as an accusation to the editor and the reviewer that there is a lack of expertise in the review since the reviewer is anonymous to the author. The author now has the choice to submit to a range of journals with policies that do not permit anonymity if they do not have faith in anonymous review.

Response: We sincerely apologize for any unintended affront, and did not wish for our comment to be negatively construed by any means. In some peer-reviewed operating grant committees, reviewers are asked to declare their level of expertise, all for the sake of aiding the peer review process. We would like to re-affirm our appreciation for your thoughtful review of our paper and the professional editorship of the PLoS One journal, and acknowledge that the debate of transparent, blinded, and double-blinded peer review is outside the scope of our paper. 

7. PLOS authors have the option to publish the peer review history of their article (what does this mean?). If published, this will include your full peer review and any attached files.

Response: We agree to publish the peer review history of our paper. 

Do you want your identity to be public for this peer review? For information about this choice, including consent withdrawal, please see our Privacy Policy.

Reviewer #1: Yes: alberto passi

Reviewer #2: No

---

## [Editor Report · Decision Letter 2]

30 Aug 2022

UDP-glucose dehydrogenase (UGDH) activity is suppressed by peroxide and promoted by PDGF in fibroblast-like synoviocytes: evidence of a redox control mechanism

PONE-D-22-03359R2

Dear Dr. Caroline D Hoemann,

We’re pleased to inform you that your manuscript has been judged scientifically suitable for publication and will be formally accepted for publication once it meets all outstanding technical requirements.

Kind regards,

Abdelwahab Omri, Pharm B, Ph.D, Laurentian University 

Academic Editor

PLOS ONE

---

## [Editor Report · Acceptance letter]

6 Sep 2022

PONE-D-22-03359R2 

UDP-glucose dehydrogenase (UGDH) activity is suppressed by peroxide and promoted by PDGF in fibroblast-like synoviocytes: evidence of a redox control mechanism 

Dear Dr. Hoemann:

I'm pleased to inform you that your manuscript has been deemed suitable for publication in PLOS ONE. Congratulations! Your manuscript is now with our production department. 

Kind regards, 

on behalf of

Dr. Abdelwahab Omri 

Academic Editor

PLOS ONE